# Code Reasoning for Software Engineering Tasks: A Survey and A Call to Action

**Saurabh Pujar**                                              *saurabh.pujar@ibm.com*
*IBM*

**Ira Ceka**                                                    *iceka@cs.columbia.edu*
*Columbia University*

**Irene Manotas**                                              *irene.manotas@ibm.com*
*IBM*

**Gail Kaiser**                                                *kaiser@cs.columbia.edu*
*Columbia University*

**Baishakhi Ray**                                              *rayb@cs.columbia.edu*
*Columbia University*

**Shyam Ramji**                                                *ramji@us.ibm.com*
*IBM*

**Reviewed on OpenReview:** *https://openreview.net/forum?id=zZa3u6LKwO*

## Abstract

The rise of large language models (LLMs) has led to dramatic improvements across a wide range of natural language tasks. Their performance on certain tasks can be further enhanced by incorporating test-time reasoning techniques. These inference-time advances have been adopted into the code domain, enabling complex software engineering (SWE) tasks such as code generation, test generation and issue resolution. However, the impact of different reasoning techniques on code-centric SWE tasks has not been systematically explored. In this work, we survey code reasoning techniques that underpin these capabilities, with a focus on test-time compute and inference-time reasoning paradigms. We examine a variety of code-specific reasoning methods and progressively build up to SWE agents, which combine planning, tool use, and multi-step interaction. We also compare the impact of different techniques on coding tasks, highlighting their relative importance and outlining open challenges and future research directions. Across commonly used models and benchmarks, we find that approaches exploiting code-specific signals (e.g., structure and execution feedback) are frequently associated with improved performance, motivating a dedicated study of code reasoning beyond natural-language reasoning. Our contributions are: (1) to the best of our knowledge, the first dedicated survey of code reasoning for SWE tasks, highlighting overarching reasoning strategies, hybrid methods, and agentic approaches; (2) a taxonomy of inference-time techniques used to drive code reasoning, accompanied by a curated set of under-explored benchmarks with high potential for SWE evaluation; (3) a comparative analysis of reasoning design patterns across commonly used models and benchmarks; and (4) a synthesis of gaps in current methods and evaluation practices, identifying under-explored areas and concrete opportunities for future research.

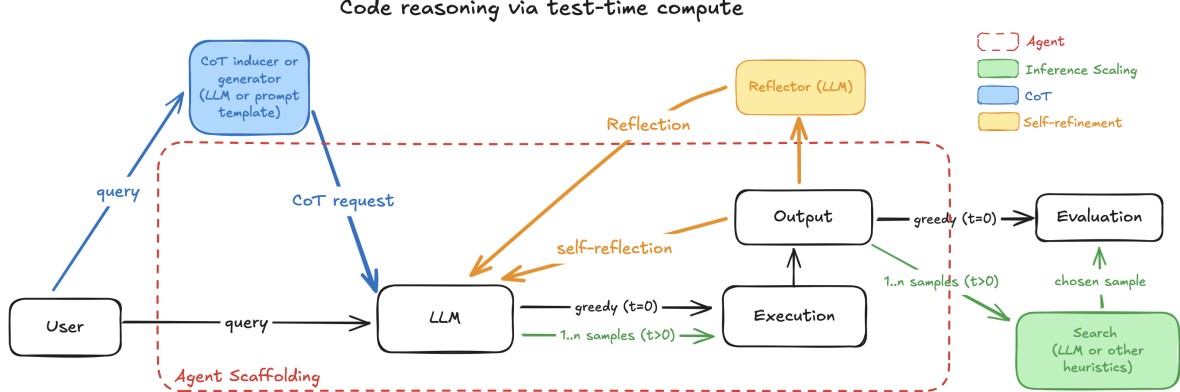

Figure 1: A simplified view of LLM inference for code tasks, illustrating both standard decoding and test-time compute-based reasoning techniques. The different colored regions indicate the core components of the reasoning methods covered in this survey. In standard inference, the user sends a query to an LLM, which produces a single greedy output (typically with temperature $t = 0$); for coding tasks, this output is executed and the resulting behavior is evaluated. CoT (Sec. 4.1) is induced by augmenting the user query with a reasoning-oriented prompt template or an auxiliary LLM. Inference Scaling (Sec. 4.3) generates multiple outputs by sampling with temperature $t > 0$, and uses search over these candidates before evaluation. Self-refine (Sec. 4.2) iteratively improves candidate outputs by having an LLM critique and revise them; the critic may be the same model (self-reflection) or a separate reflector model. Finally, an Agent (Sec. 4.4) orchestrates LLM calls and manages the execution environment, including tool use for interacting with code and external systems.

# 1 Introduction

Hindle et al., 2012 show that software is repetitive and predictable like natural language, and hence can be modeled using statistical techniques like LLMs. Subsequently, LLMs have been used effectively for a wide variety of Software Engineering (SWE) tasks[1], including code generation (Chen et al., 2021b), test generation (Mündler et al., 2025), issue resolution (Jimenez et al., 2024b) and others. Many code specific datasets (Puri et al., 2021; Khan et al., 2024), models (Li et al., 2023; Nijkamp et al., 2023) and benchmarks (Hendrycks et al., 2021a; Zhuo et al., 2025) have also been developed. Despite this progress, LLMs have been shown to be limited in their capacity to solve real-world SWE tasks, like GitHub issue resolution Jimenez et al. (2024b). Recent development of Large Reasoning Models (LRMs) Guo et al. (2025); Anthropic (2025); Jaech et al. (2024) and SWE agents have resulted in tremendous improvement on code tasks, including GitHub issue resolution.

In a recent survey, Yang et al., 2025 explore how code and reasoning reinforce each other. They compile works showing how incorporating code data improves reasoning, and how better reasoning leads to improvement on SWE tasks. Reasoning is induced in LLMs with test-time compute techniques that enable models to "think". These underlying techniques that contribute to reasoning models, include Chain-of-Thought or CoT (Wei et al., 2022b) which elicits reasoning, learning from environment feedback (Chen et al., 2024c) and exploring multiple execution paths (Yao et al., 2023a). These techniques are primarily inference time, but can include some training with model generated synthetic reasoning data as well. Many recent surveys explore reasoning techniques, SWE LLMs, benchmarks and agents, and we discuss them in Sec. 3 and summarize the topics covered in Tab. 1. We did not find any survey that explores the impact of reasoning, and specifically code-based reasoning techniques on SWE tasks.

Figure 1 illustrates a simplified implementation of test-time compute based reasoning techniques, many of which have recently been adapted to code-centric tasks. However, the literature is fragmented, with

---

[1] We use SWE tasks, Code tasks and Software engineering tasks interchangeably.

inconsistent terminology and heterogeneous evaluation protocols, making it difficult to compare approaches and identify reliable design patterns. At a high level, code reasoning methods can be grouped into chain-of-thought (Sec. 4.1), self-refinement (Sec. 4.2), and inference scaling (Sec. 4.3). Beyond these components, software engineering agents (SWE agents; Sec. 4.4) provide the scaffolding for LLMs to iteratively apply test-time compute, such as planning, tool use, execution, and revision, to solve challenging code tasks such as GitHub issue resolution. In practice, many surveyed systems combine multiple techniques, as summarized in Table 2.

To better understand the impact of these design choices, we synthesize reported results under a common set of models and benchmarks (Sec. 6). This analysis suggests that approaches that explicitly leverage properties of code, such as its syntactic structure and executability, tend to achieve stronger performance, highlighting an important distinction from reasoning in the natural language domain. We also observe that evaluation is concentrated on a small set of popular code-generation benchmarks, which limits conclusions about broader SWE capabilities. Accordingly, we catalog additional tasks and benchmarks that better reflect diverse SWE workflows (Sec. 5) and use these findings to motivate concrete directions for future research (Sec. 7).

SWE is one of the most interesting applications areas of Artificial Intelligence (AI) and there is growing research in this space. As different reasoning techniques mature and agents become more robust, it is reasonable to expect more and more SWE tasks will be automated. With our survey on code reasoning for code tasks, we hope to address this gap by making the following contributions:

**(1)** The first survey specific to reasoning for coding tasks, emphasizing reasoning techniques which borrow ideas from coding principles (Sec. 4). SWE Agents are given a special focus (Sec. 4.4) since they often rely on multiple reasoning techniques.

**(2)** A Taxonomy covering different reasoning approaches and benchmarks for code Fig. 3. We also highlight approaches employing multiple reasoning techniques for LLMs and SWE agents (Tab. 2).

**(3)** Showcase benchmarks used to study the impact of reasoning on SWE tasks. We compiled comparison tables (Tab. 5, 6, 7, 8, 9, 10, 11) showing the performance of different code reasoning and agentic approaches (Sec. 5.1). We also highlight promising benchmarks specific to code reasoning (Sec. 5.4), and surface some new agent-specific benchmarks with potential for furthering SWE research.

**(4)** Comparison and discussion of results from different reasoning techniques examined in the survey (Sec. 6). In Sec. 7, we use this discussion to motivate future work.

A categorized list of all the works covered by our survey is also available on GitHub: `https://github.com/AI4Code-IBM-Columbia/code-reasoning-for-swe-tasks`

## 2 Survey Methodology

We conducted a structured literature search using arXiv and Google Scholar to obtain broad coverage of relevant work on code reasoning for code-centric and SWE tasks. The literature search began in February 2025, with references collected from early February through May 2025. The search was updated in December 2025 to capture additional relevant work. We used advanced search queries combining terms such as *"code reasoning"*, *"reasoning"* + *"LLM"*, and *"agents"* + *"software engineering"*, *"inference scaling"*, *"reflection"*, *"refinement"*, *"code execution"* and *"code benchmarks"*. We include works that evaluate on code/SWE tasks and employ test-time (inference-time) reasoning or agentic scaffolding. Purely training-time techniques are out of scope, unless the training data includes model generated test-time reasoning data like CoT, self-reflection or agent trajectories.

To capture recent work that builds on established foundations, we also followed citation links in Google Scholar and manually screened papers that cite widely used reference works in this area. We prioritized publications from major venues (e.g., ICSE, ASE, FSE, ACL, EMNLP, ICLR, NeurIPS, and TMLR), while also including relevant arXiv preprints to reflect fast-moving developments that may not yet be represented in archival proceedings.

Code reasoning technique records range from 2022 through 2025. Records for benchmarks and tasks range from 2021 through 2025. The search queries usually returned thousands of hits, so we went through the first

few pages of the search results and screened for relevant titles. Among the records with relevant titles, we reviewed the abstract to verify whether the material was indeed relevant. If a paper was relevant, we checked the methodology to confirm that the reasoning was indeed test-time reasoning, then we checked that at least one of the tasks was related to code. From these records, we extended the search by using Google Scholar citations to include more relevant records.

To support transparency and reproducibility, we maintain an up-to-date, categorized list of surveyed papers and pointers to resources on GitHub[2]. New papers included will be categorized according to the Code Reasoning Taxonomy in Fig. 3.

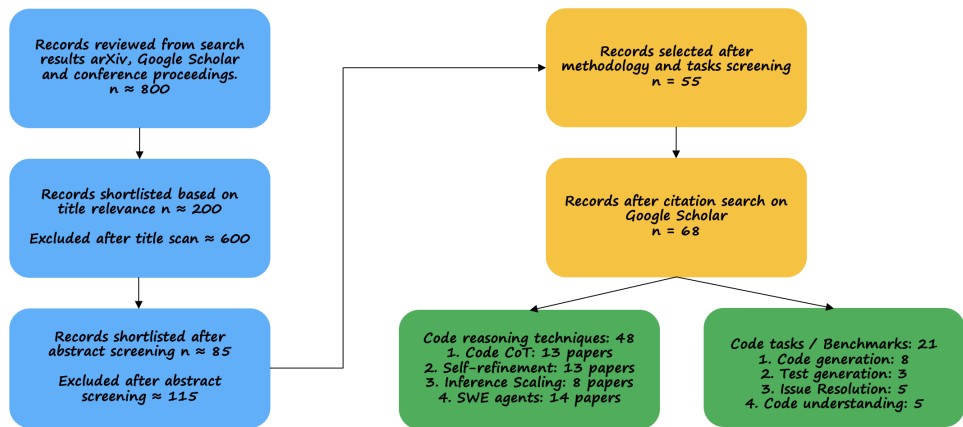

Figure 2: PRISMA-style flow of study selection and categorization. In total, 68 unique records were included. The category counts sum to 69 because one record, Wang et al. (2024c), is counted in both code reasoning techniques (CodeAct) and code tasks (M3ToolEval).

## 3 Related Surveys

Wei et al., 2022b introduce CoT as a form of in-context learning which elicits reasoning in LLMs. In the same year, Dong et al., 2022 survey in-context learning techniques and reference CoT reasoning but do not expand on it. Qiao et al., 2022 and Huang & Chang, 2022 survey methods and tasks for reasoning and extensively study CoT and other prompting approaches, but do not include software engineering tasks. Chu et al., 2023 also cover CoT reasoning extensively in a recent work. They define a more general concept of XoT or X-of-Thought, which covers concepts like Program-of-Thought Chen et al. (2022), Tree-of-Thought Yao et al. (2023a) etc. apart from CoT. However, they focus on the impact of these techniques on reasoning benchmarks while we are more interested in how reasoning impacts code specific or software engineering benchmarks. Other recent surveys also cover different types of reasoning techniques for LLMs. Xu et al., 2025 discuss reinforcement learning based reasoning techniques, but they don't discuss code specific reasoning strategies. Plaat et al., 2024 classify the in-context reasoning approaches into prompting, evaluating and control (inference scaling and search) based strategies, but they don't focus on coding tasks.

In their work titled "Code to Think, Think to Code", Yang et al., 2025 highlight the interplay between code properties and reasoning capabilities and how one enhances the other. This survey makes the case that training with code related data improves performance on Math and reasoning benchmarks, while incorporating reasoning improves performance on coding benchmarks because some code properties reinforce reasoning capabilities and vice versa. Compared to this work, we dive deeper into reasoning techniques used for coding tasks and provide a taxonomy covering different strategies. Mei et al. (2025) provide a comprehensive survey on context engineering, which involves context retrieval, processing and management. Because of the broad scope of their topic, they cover many aspects of SWE reasoning and tasks. However they have a much more

---

[2]https://github.com/AI4Code-IBM-Columbia/code-reasoning-for-swe-tasks

| Survey | Covers Reasoning | Covers SWE Tasks | Covers Agents | Provides Taxonomy | Benchmarks Coverage |
|---|---|---|---|---|---|
| Dong et al., 2022 | ✓ | ❌ | ❌ | ❌ | ❌ |
| Qiao et al., 2022 | ✅ | ❌ | ❌ | ❌ | ❌ |
| Huang & Chang, 2022 | ✅ | ❌ | ❌ | ❌ | ❌ |
| Zan et al., 2022 | ❌ | ✓ | ❌ | ❌ | ❌ |
| Chu et al., 2023 | ✓ | ✓ | ❌ | ❌ | ❌ |
| Jiang et al., 2024a | ✓ | ✓ | ❌ | ❌ | ❌ |
| Sun et al. (2024a) | ✅ | ✅ | ❌ | ❌ | ✓ |
| Plaat et al., 2024 | ✅ | ❌ | ❌ | ❌ | ❌ |
| Wang et al. (2024b) | ❌ | ✓ | ❌ | ❌ | ✓ |
| Chen et al. (2024b) | ❌ | ✅ | ❌ | ❌ | ✓ |
| Yehudai et al. (2025) | ❌ | ✅ | ✅ | ❌ | ✅ |
| Xu et al., 2025 | ✅ | ❌ | ❌ | ❌ | ❌ |
| Huynh & Lin, 2025 | ✓ | ✓ | ❌ | ❌ | ❌ |
| Yang et al., 2025 | ✓ | ✓ | ❌ | ❌ | ❌ |
| Mei et al., 2025 | ✅ | ✓ | ✓ | ✓ | ✓ |
| **Our Survey** | ✅ | ✅ | ✅ | ✅ | ✅ |

Table 1: Comparison of existing surveys along five dimensions: reasoning, SWE tasks, agents, taxonomy, and benchmark coverage. ❌ indicates that the topic is not covered at all. ✓ indicates partial coverage; for example, in the Reasoning column this may mean that only a single technique such as CoT is discussed, and in the SWE tasks column that only code generation is considered. ✅ indicates comprehensive, in-depth coverage, specifically for code and SWE tasks. Our survey provides in-depth coverage across all these dimensions for test-time compute-based code reasoning on SWE tasks.

general focus because of which they are unable to cover many aspects of code specific reasoning and different code related benchmarks and tasks.

A lot of surveys do cover impact of LLMs and Agents on Software Engineering tasks but none so far have focused on reasoning based strategies. Zan et al., 2022 survey 27 LLMs for natural language to code generation task. Jiang et al., 2024a undertake an extensive survey for code generation covering not just LLMs but also LLM architectures, many different research topics, benchmarks and datasets, encompassing a total of 235 papers. Sun et al. (2024a) also do a wide ranging survey covering 50 different models and their variants along with 20 different code-related task categories. Huynh & Lin (2025) survey many topics in this space including challenges and applications. Apart from surveys covering multiple topics from the domain of AI for code/software engineering, there are also surveys that are more topic specific. Wang et al., 2024b focus exclusively on reinforcement learning in code generation. Chen et al., 2024b survey different evaluation techniques for coding tasks. Yehudai et al., 2025 also focus on evaluation, but of LLM-agents and including Software Engineering (SWE) Agents.

We did not find any survey specific to code based reasoning techniques for software engineering tasks, covering agents and benchmarks and including a taxonomy.

## 4 Code Reasoning: Techniques

Brown et al., 2020 show that LLMs are few-shot learners. Performance of LLMs on reasoning tasks is further enhanced by a certain kind of prompting called Chain-of-thought or CoT (Wei et al., 2022b) prompting which elicits LLM reasoning. Wei et al., 2022a suggest that in-context learning ability of LLMs, including CoT reasoning, is an emergent property of LLMs. Code CoT papers (Li et al., 2025b; Jiang et al., 2024b; Pan & Zhang, 2025 and others) suggest that code reasoning is a specific kind of reasoning and CoT can be more impactful when induced with prompts that recognize this difference. We survey such techniques in Sec. 4.1.

One way code output is different from natural language output is that it can be executed and tested to validate its correctness. Self-refinement techniques using code execution as a way to evaluate and improve LLMs' output, such as Self-debugging (Chen et al., 2024c), CodeCoT (Huang et al., 2023), LEVER (Ni et al., 2023) and others, are covered in Sec. 4.2.

Yao et al., 2023a state that "System 2" thinking should involve exploring diverse solution paths rather than greedily picking one. They connect CoT with inference scaling to enable exploration of multiple reasoning

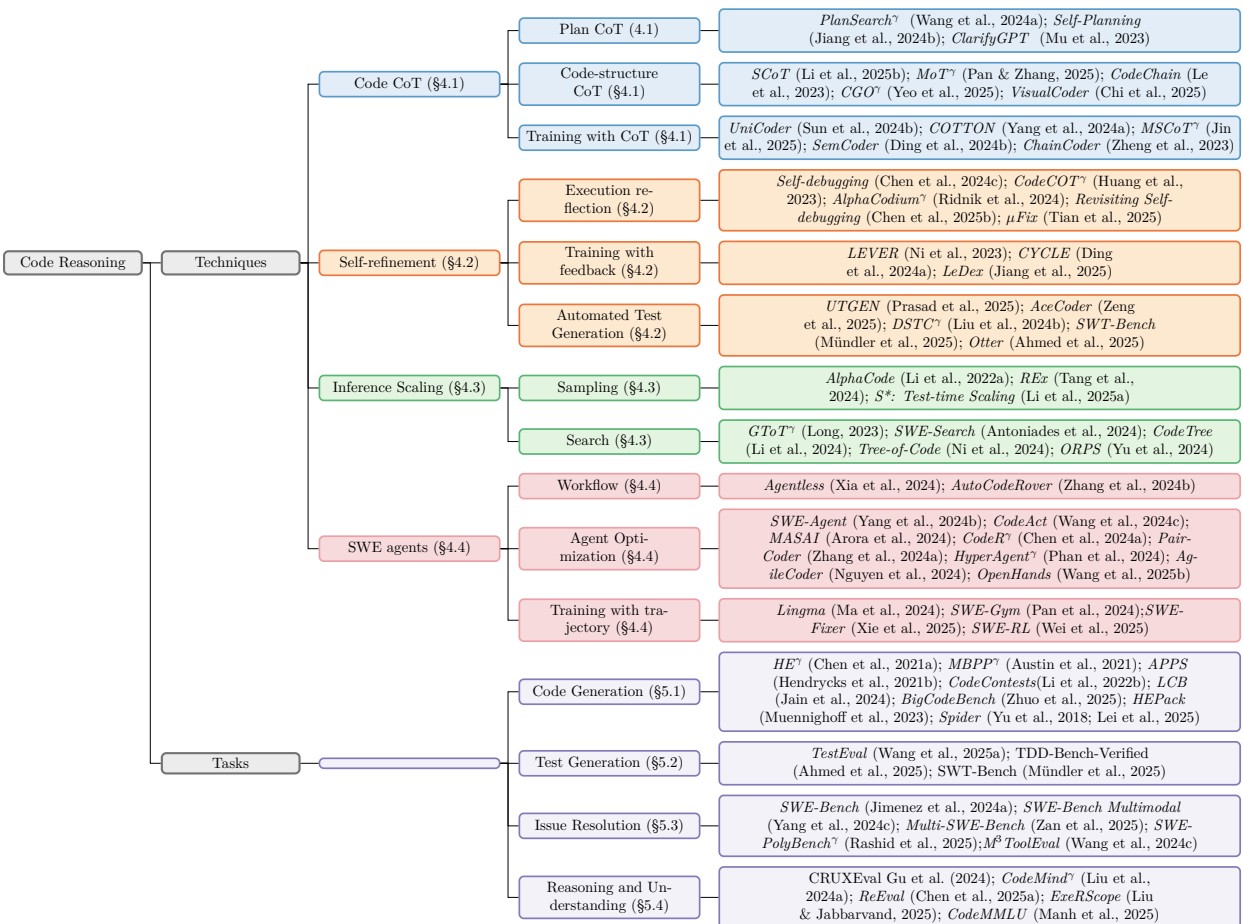

Figure 3: Code Reasoning Taxonomy. We organize prior work on code reasoning along two axes: *techniques* and *tasks*. Techniques are methods that elicit, enhance, or exploit reasoning in LLMs. Tasks (often instantiated as benchmarks) are used to evaluate an LLM's code-reasoning capability. Techniques are frequently combined within a single system; in this taxonomy, we categorize each publication by its dominant technique (i.e., the primary mechanism emphasized by the method). A [γ] symbol right after the technique's name indicates the publication has not been published yet in a peer-reviewed venue or is under revision at the time of this publication. Table 2 highlights works that explicitly integrate multiple techniques.

paths. Inference scaling involves setting a temperature greater than 0 to generate multiple candidate solutions or samples during inference or test time, and then picking the best candidate. Li et al., 2022a effectively leverage this technique to generate competition level code. Sec. 4.3 covers inference scaling used to explore multiple reasoning paths for software engineering tasks.

Many approaches use a combination of these techniques, although one technique usually dominates. The dominant technique is the core method, contribution or novelty of a publication. Tab. 2 shows approaches which rely on multiple techniques.

## 4.1 Code Chain-of-thought

CoT prompts for code can be categorized as plan-based or structure based. Plan-based CoT is a natural language articulation of steps that need to be taken to solve a coding problem. Code-structure-based CoT utilizes some code structure or programming concepts. Besides these two prompting only techniques, another approach used by many is fine-tuning or instruction tuning for software engineering tasks with code CoT data.

**Plan CoT.** Several recent approaches enhance code generation by explicitly modeling intermediate reasoning or problem understanding steps. For instance, `PlanSearch` Wang et al. (2024a) generates 3–6 problem observations, combines them into natural language plans, and translates these into pseudocode and then code. `Self-Planning` Jiang et al. (2024b) uses few-shot prompting to extract a high-level plan from the problem, which guides code generation. `ClarifyGPT` Mu et al. (2023) employs test generation to construct clarifying questions and answers that are appended to the prompt for code synthesis.

**Code-structure CoT.** In `SCoT`, Li et al., 2025b use programming structures, like sequence, branch and loop, as steps towards intermediate code, which is used to prompt the model to generate code. `Chain of grounded objectives (CGO)` Yeo et al. (2025) embed appropriately structured functional objectives into the input prompts to enhance code generation. Pan & Zhang, 2025 propose a novel prompting technique, `Modularization-of-thought (MoT)`, which exploits modularization principles to decompose complex programming problems into smaller independent reasoning steps, via a multi-level reasoning graph. Le et al., 2023 also elicit modularized code generation but in a multi-step technique called `CodeChain`, which is a chain of self-revisions applied by picking potentially correct representative sub-modules. Recently, VisualCoder Chi et al. (2025) integrated multimodal CoT reasoning with a visual Control Flow Graph (CFG) to obtain deeper insights into execution flows and improve code execution reasoning and program repair generated solutions.

**Training with CoT.** Sun et al., 2024b define `UniCoder`; they use an intermediate representation CoT based on PL conventions and use this to instruction-tune a model on a multi-task learning objective. Yang et al., 2024a generate high-quality CoTs based on the `COTTON` framework, which trains light-LMs ($< 10$B parameters) to generate CoT comparable to those generated by strong teacher LLMs. `ChainCoder` Zheng et al. (2023) generates code iteratively in a "course-to-fine" approach and trains a model using an AST-based vocabulary. `SemCoder` Ding et al. (2024b) uses a monologue reasoning approach to train a model to learn program semantics, which is generated by asking the Code LLM to summarize the program functionalities, key properties and constraints, and reason about code execution step-by-step using a bi-directional monologue reasoning method. `MSCoT` Jin et al. (2025) extends SCoT Li et al. (2025b) to 11 more programming languages beyond Python; a trained MSCoT model generates structured-CoT before producing code in multiple languages.

## 4.2 Self-refinement

Self-refinement involves executing LLM-generated code in an environment and having the same or a different LLM reason about the execution output. This reasoning can be fed back to the LLM to refine the code.

**Execution reflection.** These strategies utilize code execution feedback to select the final prediction from a LLM. In Chen et al. (2024c), the `Self-debugging` approach, teaches the model to self-debug i.e., debug the model's predicted code, via few shot prompting and without additional model training. A similar approach was taken in Code Chain-of-Thought (`CodeCoT`) by Huang et al. (2023), where CoT is used as a first step to generate the code, then a LLM generates test cases to validate whether the code has syntax errors during the execution. `AlphaCodium`, proposed by Ridnik et al. (2024), is a flow with two key phases, (a) pre-processing to generate reflection and (b) iterative code generation, to improve code LLM performance that does not require training a model. In `revisited self-debugging` Chen et al. (2025b) authors explored both post-execution and in-execution self-debugging, leveraging self-generated tests. More recently, Tian et al. (2025) proposed $\mu$Fix (Misunderstanding Fixing) where thought-eliciting prompting techniques are combined with feedback-based prompting to improve the code generation performance of LLMs.

**Training with feedback.** We pinpoint approaches that train an LLM, leveraging execution data, to improve model performance. LEarning to VERify Ni et al. (2023) (`LEVER`) is an approach where verifiers are trained to check whether the generated code is correct or not based on three sources of information: the natural language input, the program itself, and its execution results. CYCLE Ding et al. (2024a) trains code LLMs to self-refine using natural language specifications, generated code, and execution feedback, while avoiding repeated errors via a Past Generation Mask. Similarly, Jiang et al. (2025) proposed `LEDEX`, a training framework to improve the self-debugging capability of LLMs using a chain of explanations on the wrong code followed by code refinement.

**Automated Test Generation.** Unit Tests (UT) are one of the fundamental pieces to assess the correctness of code and give execution-based feedback to code generation models. UTGEN Prasad et al. (2025) is a data creation and training recipe that bootstraps training data for UT generation and works by perturbing code to simulate errors, generating failing tests and augmenting it with CoT rationales.

AceCoder Zeng et al. (2025) leverages automated large-scale test-case synthesis to enhance code model training. They proposed a pipeline that generates extensive *(question, test-cases)* pairs from existing code data. Similarly, Liu et al. (2024b) propose Direct Preference Learning with Only Self-Generated Tests and Code (DSTC), using only self-generated code snippets and tests to construct preference pairs with direct preference learning to improve LM coding accuracy without external annotations. ASTER Pan et al. (2025a) is a multilingual UT-generator built with LLMs guided by lightweight program analysis.

## 4.3 Inference Scaling

Several approaches to code generation, code repair, and test-case generation use *tree-based* strategies to guide decisions and explore reasoning paths, while others use sampling.

**Sampling.** In `AlphaCode`, Li et al. (2022a) filter and cluster samples according to program behavior on model-generated test inputs, selecting one candidate per cluster. The authors of `REx` (Tang et al., 2024) frame iterative code repair, or *refinement*, as a multi-armed bandit problem which is solved using Thompson sampling. In `S*`, Li et al. (2025a) take a hybrid sampling approach, first generating N diverse programs in parallel then refining them using iterative debugging (informed by execution). CodeTree (Li et al., 2024) and ToC (Ni et al., 2024) both model reasoning as tree search—CodeTree combines planning, execution-guided reasoning, and heuristics (test-pass rate, LM critique) via multi-agent roles, while ToC uses a binary pass/fail heuristic with reflective, multi-strategy execution for diverse solutions.

**Search.** `Tree-of-Thoughts (ToT)` (Yao et al., 2023a) allows LMs to explore multiple reasoning paths over thoughts, where thoughts are language sequences that serve as intermediate steps towards problem solutions and represent the states or nodes of the tree. Similarly, `Guided tree-of-thought (GToT)` (Long, 2023) uses tree-search guided by an LLM heuristic; it generates intermediate solutions through prompting, employs a checker to validate these solutions, and uses a controller to manage search and backtracking, enabling long-range reasoning. For test generation, Ouédraogo et al. (2024) show that `GToT` effectively produces syntactically-correct, compilable test suites with superior code coverage. Yu et al. (2024) propose `Outcome-Refining Process Supervision (ORPS)`, a beam-search approach for code generation over a "reasoning tree". `SWE-Search` (Antoniades et al., 2024) is a moatless-tools (Orwall, 2024) based multi-agent framework which integrates Monte-Carlo Tree Search with self-improvement for bug-fixing.

## 4.4 SWE agents

Agentic systems use many of the reasoning techniques described in Sec. 4 for different tasks. Software Engineering (SWE) agents take a programming problem and iteratively solve it by self-debugging based on the feedback provided by the environment. The self-debugging is enabled by CoT style natural language reflection (Shinn et al., 2023) on environment feedback. The reasoning is done by an LLM which interacts with the agent execution environment with tool calls (Yao et al., 2023b).

**Workflow.** Schluntz & Zhang, 2024 draw a distinction between Agents and LLM-based workflows stating that the latter are simpler, have a fixed path and do not require an LLM to make a decision. Agentless (Xia et al., 2024) is a three step process for Github issue resolution involving localization, repair and patch validation. AutoCodeRover (Zhang et al., 2024b) uses program structure, in the form of an Abstract Syntax Tree (AST), to enhance code search and look at a software as classes and functions, rather than a set of files.

**Agent Optimization** can often lead to performance gains. There can be many ways to improve an SE agent, including but not limited to, better environment management or agent-environment interface, improved workflow or architecture, and incorporating more tools. SWE-Agent (Yang et al., 2024b) is an agent capable of editing repository-level code by generating a thought and a command, and subsequently incorporating the feedback from the command's execution into the environment. In CodeAct, Wang et al. (2024c) propose to use executable Python code to consolidate LLM agents' actions into a unified action space.

| Approach | Code CoT | | | Self-refinement | | | Inference Scaling | | SWE agents | | |
|---|---|---|---|---|---|---|---|---|---|---|---|
| | Plan | Struct | Train CoT | Exec. ref. | Train feedback | ATG | Samp. | Search | Work-flow | Agent Opt. | Train traj. |
| AlphaCode (2022a) | | | | | | | ✓ | ✓ | | | |
| ClarifyGPT (2023), Self-Planning (2024b) | ✓ | | | | | | | | | | |
| CodeChain (2023), SCoT (2025b), CGO (2025), MoT (2025), VisualCoder (2025) | | ✓ | | | | | | | | | |
| ChainCoder (2023), UniCoder (2024b), MSCoT (2025) | | ✓ | ✓ | | | | | | | | |
| LEVER (2023), Self-Debugging (2024c), ORPS (2024), REx (2024) | | | | ✓ | | | ✓ | ✓ | | | |
| CodeCoT (2023), AlphaCodium (2024) | ✓ | | | ✓ | | ✓ | | | | | |
| GToT (2023) | | | | | | | ✓ | ✓ | | | ✓ |
| PlanSearch (2024a) | ✓ | | | ✓ | | | ✓ | | | | |
| COTTON (2024a) | ✓ | | ✓ | | | | | | | | |
| SemCoder (2024b) | | ✓ | ✓ | | | ✓ | | | | | |
| CYCLE (2024a) | | | | ✓ | ✓ | | ✓ | | | | |
| DSTC (2024b) | | | | | ✓ | ✓ | | | | | |
| CodeTree (2024), Tree-of-Code (2024), SWE-Search (2024) | | | | | | | ✓ | ✓ | | ✓ | |
| Agentless (2024) | | | | ✓ | | | | | ✓ | | |
| AutoCodeRover (2024b), PairCoder (2024a) | | | | ✓ | | | | | ✓ | ✓ | |
| CodeAct (2024c) | | | | ✓ | | | | | | ✓ | ✓ |
| OpenHands (2025b), MASAI (2024), CodeR (2024a) | | | | ✓ | | | | | | ✓ | |
| AgileCoder (2024), HyperAgent (2024), SWE-Agent (2024b) | | | | ✓ | | | | | | ✓ | |
| Lingma (2024), SWE-Fixer (2025) | | | | ✓ | | | | | ✓ | | ✓ |
| SWE-Gym (2024) | | | | ✓ | | | ✓ | ✓ | | ✓ | ✓ |
| Rev. Self-Debugging (2025b) | | | | ✓ | ✓ | | | | ✓ | | |
| S* (2025a) | | | | ✓ | ✓ | ✓ | | | | ✓ | |
| µFix (2025) | ✓ | | | ✓ | | | | | | | |
| LeDex (2025) | ✓ | | | ✓ | ✓ | | ✓ | | | | |
| UTGEN (2025), AceCoder (2025) | | | | | ✓ | ✓ | ✓ | | | | |
| SWT-Bench (2025), Otter (2025) | ✓ | | | | | ✓ | ✓ | | | ✓ | |
| SWE-RL (2025) | | | | | | | | | | | ✓ |

Table 2: Summary of the test-time compute-based reasoning techniques used in the papers surveyed. For each work, we assign a dominant technique as shown in the taxonomy Fig. 3. Additional techniques used by the same approach are marked with ✓. For example, PlanSearch is categorized under Code CoT as its dominant technique, but it also incorporates elements of Self-refinement and Inference Scaling.

OpenHands (Wang et al., 2025b) is a platform for developing flexible AI agents that interact with the digital world by writing code, interacting with the command line or browsing the web. This platform allows for integration of other specialist agents, like CodeAct (Wang et al., 2024c) for software engineering. There are other multi-agent techniques like MASAI Arora et al. (2024), CodeR Chen et al. (2024a), PairCoder Zhang et al. (2024a), HyperAgent Phan et al. (2024) and AgileCoder Nguyen et al. (2024).

**Training with trajectory.** Some agentic systems improve the underlying reasoning model by training on agent trajectories, which include steps like CoT, tool calls, and patches. Ma et al. (2024) note that software evolution spans code, reasoning, tools, and cross-role interactions, and fine-tune their `Lingma` SWE-GPT models (7B, 72B) on repository understanding, bug localization, patching, and rejection sampling from pull requests. Pan et al., 2024 build `SWE-Gym` from 2,438 real-world Python tasks—each with a runnable codebase, unit tests, and an NL spec. Using OpenHands scaffolding (Wang et al., 2025b), they fine-tune Qwen2.5-Coder-32B (Hui et al., 2024) on 491 agent–environment trajectories and train a verifier on the same data for scalable inference. `SWE-Fixer` (Xie et al., 2025) uses a fine-tuned Qwen2.5-7B retriever, boosted with BM25, to identify relevant files, while a fine-tuned Qwen2.5-72B editor generates patches for GitHub issues. In `SWE-RL` (Wei et al., 2025), Llama 3 (Grattafiori et al., 2024) is trained with lightweight rule rewards and GRPO (Shao et al., 2024) on 11M filtered PRs, producing Llama3-SWE-RL-70B, the top medium-sized model on SWE-bench Verified (OpenAI, 2024b) upon release.

## 5    Code Reasoning: Tasks

In this section we discuss the different tasks and benchmarks which are used to evaluate code reasoning techniques described in Sec. 4.

### 5.1    Code Generation

For code generation, a popular task, most common benchmarks include *HumanEval (HE)* (Chen et al., 2021a), `HumanEvalPack` (Muennighoff et al., 2023), *MBPP* (Austin et al., 2021), *APPS* (Hendrycks et al., 2021b), and *CodeContests* (Li et al., 2022b).

More recently, *LiveCodeBench (LCB)* (Jain et al., 2024) collected new problems over time from contests platforms including LeetCode, AtCoder, and CodeForces. *BigCodeBench* (Zhuo et al., 2025) challenges LLMs to invoke multiple function calls as tools from multiple libraries and domains for different fine-grained tasks. CRUXEval (Gu et al., 2024) includes both input and output predictions to evaluate code reasoning and code execution, respectively. ConvCodeBench (Han et al., 2025) is a benchmark for interactive code generation, it uses pre-generated feedback logs, avoiding costly LLM calls for verbal feedback while maintaining strong correlation with live results; *Spider* (Yu et al., 2018; Lei et al., 2025) is a benchmark to evaluate the generation of SQL queries from natural language.

### 5.2    Test Generation

For test generation, benchmarks like *TestEval* (Wang et al., 2025a) can help on three different aspects: overall coverage, targeted line/branch coverage, and targeted path coverage. *SWT-Bench* (Mündler et al., 2025) is another github based test-generation benchmark; Otter (Ahmed et al., 2025) too, proposed an LLM-based solution to generate test cases from issues. [3]

### 5.3    Issue Resolution

For GitHub issue resolution, *SWE-Bench* (Jimenez et al., 2024a) is a popular benchmark. Other variations of SWE-Bench include: *SWE-Bench Multimodal* (Yang et al., 2024c) for visual and user-facing components, and *Multi-SWE-Bench* (Zan et al., 2025) and *SWE-PolyBench* (Rashid et al., 2025) for more programming languages besides Python. $M^3$ *ToolEval* (Wang et al., 2024c) is used for multi-turn, multi-tool complex tasks.

---

[3]Appendix A.3 lists metrics that can be used to assess code LLM performance.

## 5.4 Reasoning and Understanding

Evaluating the ability of LLMs to both correctly and soundly reason about runtime behavior of code can help understand and verify whether the generated code aligns with the intended goal. *ReEval* (Chen et al., 2025a) helps to analyze how Code LLMs reason about runtime behaviors (e.g., program state, execution paths) of programs. ExeRScope (Liu & Jabbarvand, 2025) analyzes the code execution reasoning output from LLMs for different code reasoning benchmarks and helps understand the impact of code properties (program constructs, complexity, dynamic program properties, and variable types) for such benchmarks. *CodeMMLU* (Manh et al., 2025) is a large benchmark to evaluate both code understanding and code reasoning through a multiple-choice question-answering approach. CodeMind (Liu et al., 2024a) is a code reasoning benchmark for LLMs, evaluating Independent Execution Reasoning (IER), Dependent Execution Reasoning (DER), and Specification Reasoning (SR) tasks and metrics.

## 6 Comparison and Discussion

The ideal way to understand the impact of different code reasoning techniques is an exhaustive comparative study under a common experimental setup. However, such a controlled evaluation is often impractical. To enable fairer comparisons across heterogeneous experimental settings, we synthesize reported results on coding benchmarks while conditioning on the same underlying LLMs (i.e., comparing techniques evaluated with the same model). Most of the comparable results are on code generation benchmarks, with a few on issue resolution. We summarize reported performance for the techniques described in Sec. 4 on the benchmarks and tasks in Sec. 5. We further visualize technique performance on the intersection of models and benchmarks for which results are available, making cross-technique trends easier to interpret. This analysis is not exhaustive; it is restricted to benchmarks and models shared by only a subset of techniques. All comparisons are cross-paper; each result is taken from its respective original paper. We do not normalize for differences in prompts, execution harnesses, or evaluation details. The complete set of results are in the appendix in Tables 5, 6, 7, 8, 9, 10, and 11. Observations 1–5 are scoped to code generation tasks, where the majority of comparable results are available. Observation 6 extends to issue resolution, supported by SWE-Bench results. We report the margins of difference for all comparisons in Tab. 3. We discuss threats to the validity of this cross-paper comparison approach in Sec. 8.1.

## 6.1 Plan CoT vs. Code Structure CoT

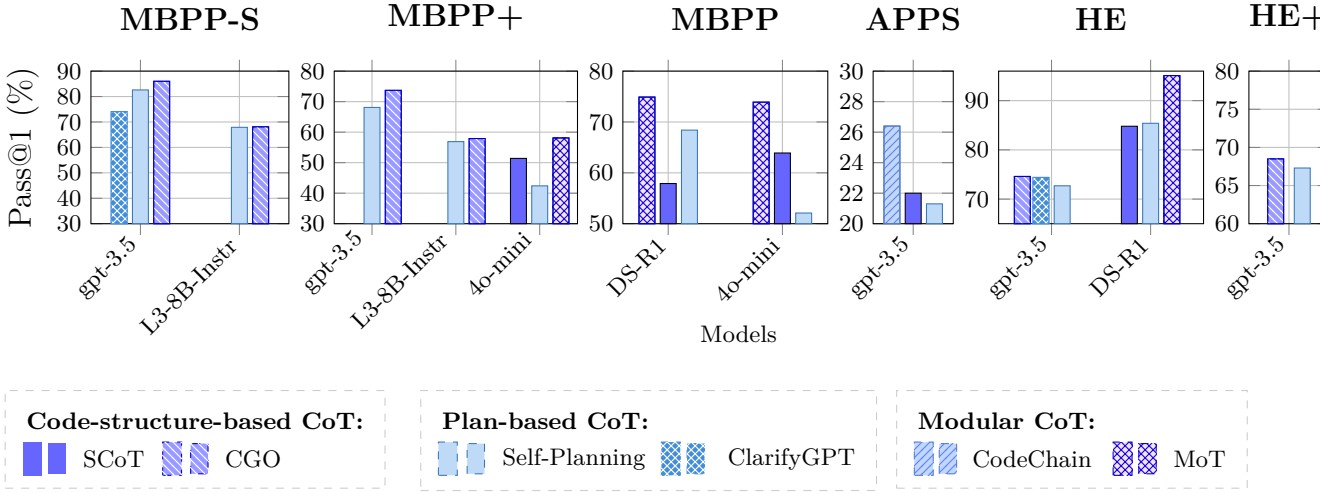

Figure 4: Code Structure CoT vs. planning-based CoT on benchmarks MBPP-S, MBPP+, MBPP, APPS, HE, and HE+ with models GPT-3.5-turbo (GPT-3.5), Llama3-8B-Instruct (L3-8B-Instr), GPT-4o mini (4o-mini) and DeepSeek-R1 (DS-R1). Modular CoT is a sub-category of Structure-aware CoT. Note: Y-axis scales differ across subplots.

Section 4.1 surveys works on code-specific CoT prompting and categorizes them into plan-based and structure-based. Fig. 4 shows the performance of different plan and structure based techniques on code generation benchmarks (Sec. 5.1). Structure-aware CoT techniques tend to outperform plan-based approaches across multiple code generation benchmarks. With gpt-3.5-turbo, CGO outperforms Self-Planning on MBPP-S, MBPP+, HE, and HE+, and surpasses ClarifyGPT on MBPP-S and HE. This pattern extends to Llama-3-8B-Instr, where CGO achieves better performance than Self-Planning on MBPP-S and MBPP+. With gpt-4o mini, SCoT demonstrates superior performance over Self-Planning on MBPP and MBPP+, and on APPS with gpt-3.5-turbo. Similarly, CodeChain outperforms Self-Planning on APPS with gpt-3.5-turbo. These results suggest that code structure-based techniques tend to outperform plan-based approaches in the reported code generation benchmarks.

> **Observation 1: Reported results across several studies suggest that structure-aware CoT strategies can outperform planning-based CoT strategies in some code generation benchmarks.**

## 6.2 Modular Code Structure CoT

Code structure aware CoT can be sub-categorized into modular approaches, as shown in Fig. 4. Modularity improves upon structure-based CoT by providing ultra-localized scoping with more clearly defined and specific functionality, eliminating the chance of error propagating to subsequent steps. MoT and CodeChain represent modular CoT techniques, more specific types of structure-aware prompting. MoT demonstrates superior performance over SCoT and Self-Planning with DeepSeek-R1 (Guo et al., 2025) on MBPP and HE, with comparable results on MBPP and MBPP+ using gpt-4o mini. CodeChain exhibits similar performance gains, outperforming SCoT and Self-Planning on APPS with gpt-3.5. Building on Observation 1, these results suggest that modular formats can outperform other structure-aware Code CoT prompting techniques on the code generation benchmarks studied.

> **Observation 2: Reported results across several studies suggest that modular CoT techniques can outperform other structure-aware and plan-based CoT approaches in some code generation benchmarks.**

## 6.3 Self-refinement vs. Code CoT

Self-refinement (Sec. 4.2) uses the execution output as feedback to iteratively improve the model generated code. Incorrect intermediate code is discarded and the model can improve upon it. As can be seen in Fig. 5, using self-refinement has a bigger impact on the studied code generation benchmarks than any code CoT based technique.

Across multiple code generation benchmarks, self-refinement methods tend to outperform CoT approaches. With gpt-3.5, $\mu$Fix and Self-Debugging surpass other CoT baselines (CGO, SCoT, Self-Planning, ClarifyGPT) and outperform UniCoder on HumanEval. On MBPP-ET with gpt-3.5, these methods also outperform ClarifyGPT, while Self-Debugging achieves particularly strong gains on MBPP with gpt-3.5, surpassing SCoT by a large margin. With Claude-3.5 (Anthropic, 2024), Revisiting Self-Debugging beats PlanSearch on HE+. Similar patterns emerge on APPS with gpt-3.5, where $\mu$Fix outperforms CodeChain, SCoT, and Self-Planning. These findings extend to DeepSeek-Coder (Guo et al., 2024), where $\mu$Fix and Self-Debugging outperform SCoT, and CYCLE models, which are smaller finetuned models, also surpass SCoT.

> **Observation 3: Reported results across several studies suggest that execution-aware strategies can outperform Code CoT based methods in some code generation benchmarks.**

## 6.4 Inference Scaling

Inference Scaling (Sec. 4.3) involves generating multiple LLM outputs and then selecting the best candidate among them for evaluation. In Fig. 6, we can see that inference scaling outperforms code CoT on the code generation benchmarks studied. ORPS outperforms MoT, SCoT, and Self-Planning on MBPP with GPT-4o mini (OpenAI, 2024a). REx with GPT-4 (Achiam et al., 2023) also claims to achieve the state-of-the-art on

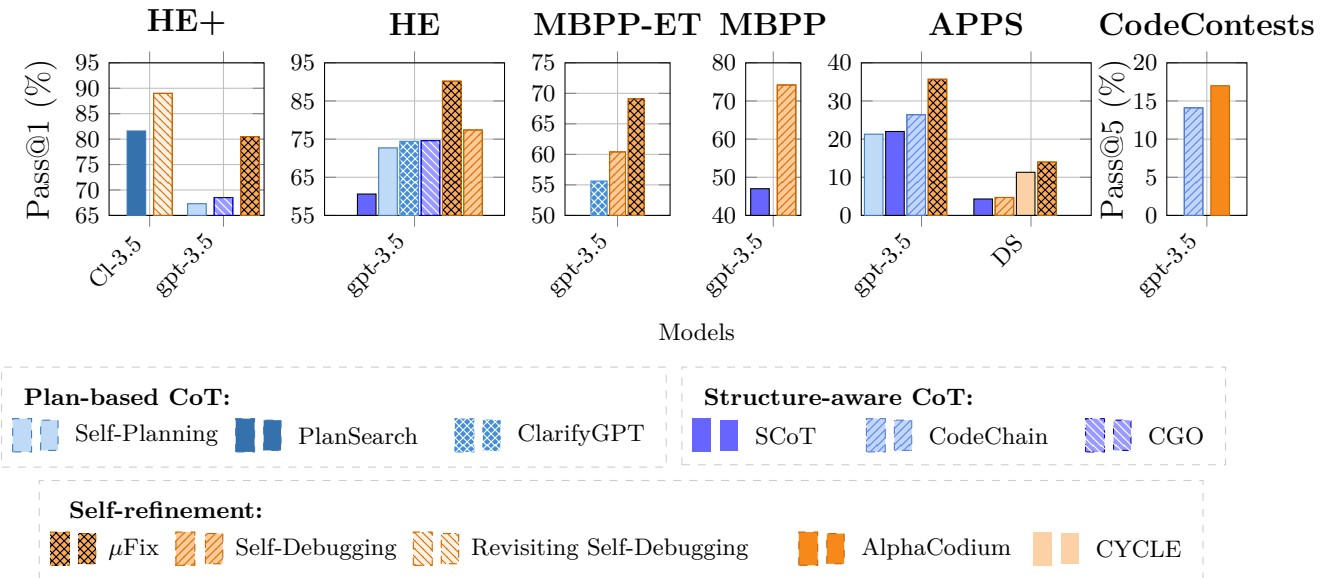

Figure 5: Comparison between code CoT and self-refinement techniques on code generation benchmarks HE+, HE, MBPP-ET, MBPP, APPS, and CodeContests with models Claude 3.5 Sonnet (Cl-3.5), GPT-3.5-turbo (GPT-3.5), and DeepSeek-Coder (DS). Code CoT is sub-categorized into plan-based and structure-aware CoT. CodeContests results are reported as pass@5. Note: Y-axis scales differ across subplots.

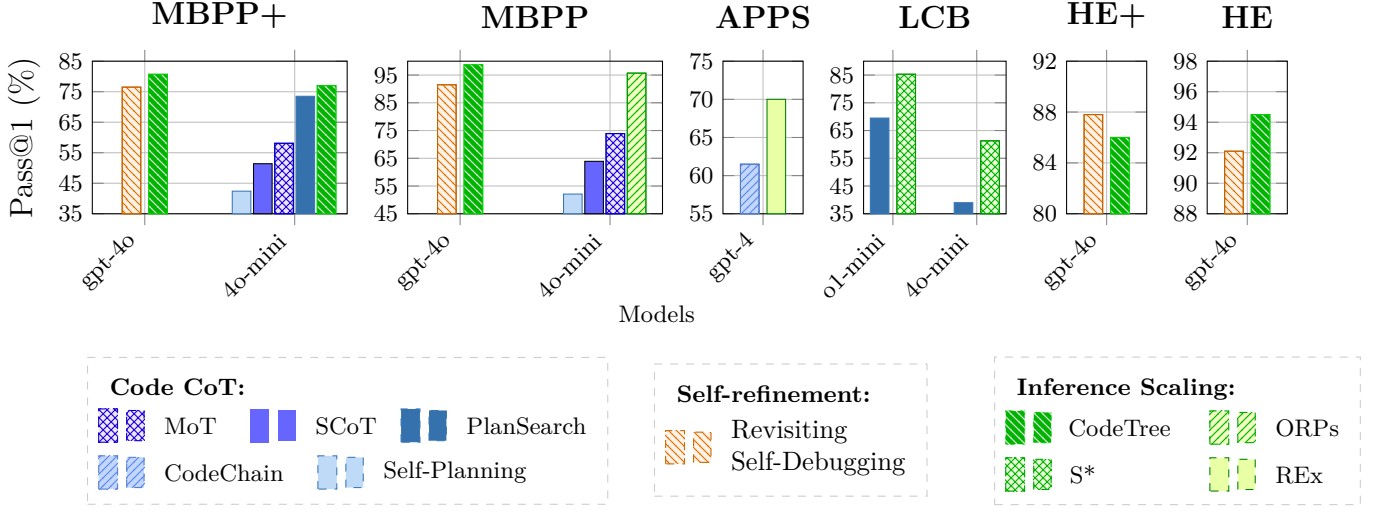

Figure 6: Performance comparison between Inference Scaling, Code CoT and Self-refinement on Code Generation benchmarks of MBPP+, MBPP, APPS, LCB, HumanEval+ (HE+) and HumanEval (HE) with models GPT-4o, GPT-4o mini (4o-mini), GPT-4 and OpenAI O1-mini (o1-mini). Note: Y-axis scales differ across subplots.

APPS, with roughly 70%, beating CodeChain. S* also beats PlanSearch on LCB with o1-mini (Jaech et al., 2024) and GPT-4o mini. Interesting to note PlanSearch, which incorporates inference scaling techniques as highlighted in Tab. 2, outperforms other Code CoT methods by a big margin on MBPP+ with GPT-4o mini. CodeTree also does better than multiple Code CoT methods on MBPP+ with GPT-4o mini.

While there is some evidence that inference scaling could outperform self-refinement, as CodeTree outperforms Revisiting Self-Debugging on MBPP+ (80.7 vs. 76.5), MBPP (98.7 vs. 91.5), and HumanEval (94.5 vs. 92.1), Revisiting Self-Debugging performs better on HumanEval+ (87.8 vs. 86.0). Since these comparisons are limited to a single inference scaling and self-refinement approach pair, this remains an open question.

> **Observation 4: Reported results across several studies suggest that approaches integrating inference scaling can outperform CoT-dominant strategies in some code generation benchmarks.**

## 6.5 SWE agents

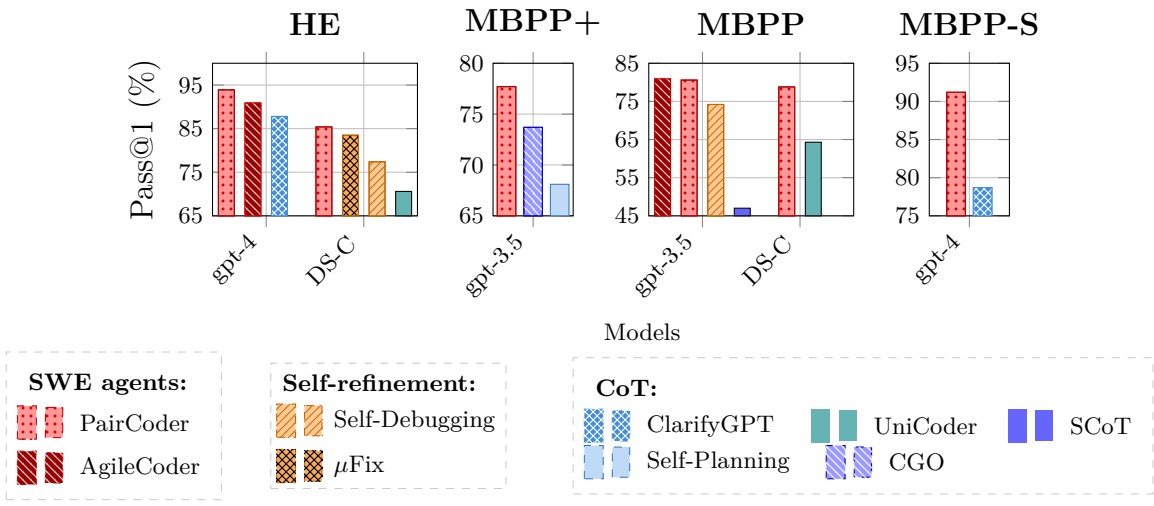

Figure 7: Comparison of SWE agents-based approaches with self-refinement and code CoT-based approaches on code generation benchmarks of HE, MBPP+, MBPP, MBPP-S with models GPT-4, DeepSeek-Coder (DS-C) and GPT-3.5-turbo (GPT-3.5). Note: Y-axis scales differ across subplots.

SWE agents (Sec. 4.4) succeed by integrating chain-of-thought reasoning, execution-based validation, and sampling into a unified framework–thus leveraging code's structured syntax, executable semantics, and error feedback all in one. Fig. 7 shows that agent-based approaches perform better than self-refinement and code CoT on the code generation benchmarks studied.

Agentic approaches tend to outperform both CoT and self-refinement methods across the benchmarks and models studied. With GPT-4, PairCoder and AgileCoder outperform ClarifyGPT on HumanEval, and Pair-Coder surpasses ClarifyGPT on MBPP-S. On GPT-3.5, PairCoder beats CGO and Self-Planning on MBPP+, while both PairCoder and AgileCoder surpass SCoT and Self-Debugging on MBPP. With DeepSeek-Coder, PairCoder outperforms $\mu$Fix, Self-Debugging, and UniCoder on HumanEval, and also UniCoder on MBPP.

> **Observation 5: Reported results across several studies suggest that orchestrating multiple reasoning techniques through agentic scaffolding can outperform single-strategy approaches in some code generation benchmarks.**

## 6.6 SWE agents with Inference Scaling

SWE agents generally have some elements of CoT and self-refinement as part of the scaffolding. Techniques that combine inference scaling with SWE agents have also been proposed. Fig. 8 shows that such techniques

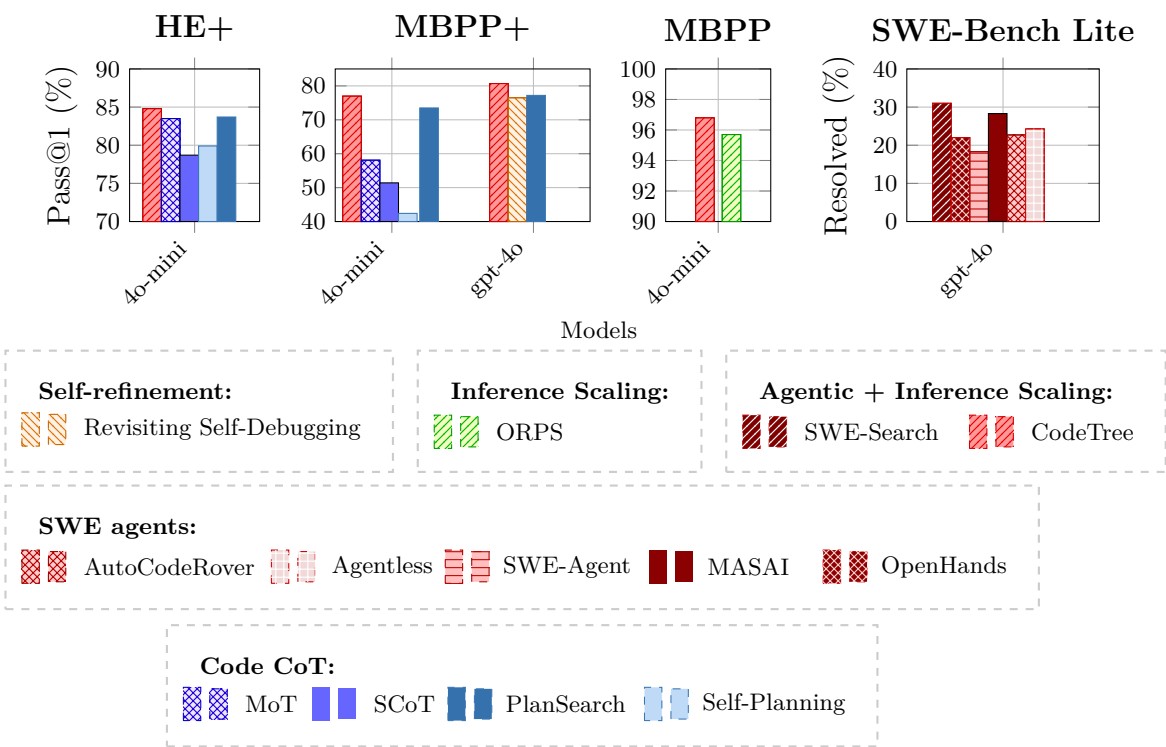

Figure 8: Performance comparison of different techniques on code generation benchmarks like HE+, MBPP+, MBPP (pass@1) and GitHub issue resolution benchmark of SWE-Bench Lite (resolved rate). The models used are GPT-4o mini (4o-mini) and GPT-4o. Note: Y-axis scales differ across subplots.

which combine inference scaling with SWE agents perform better than SWE agents, self-refinement and inference scaling based approaches on the code generation and issue resolution benchmarks studied.

Across multiple benchmarks shown, inference scaling integrated with agentic approaches demonstrates strong performance. With GPT-4o mini, CodeTree outperforms MoT, SCoT, Self-Planning, and PlanSearch on HumanEval+ and MBPP+, and surpasses ORPS on MBPP. These gains extend to GPT-4o, where CodeTree similarly outperforms these CoT-based strategies. SWE-Search, which combines inference scaling within an agentic framework, dominates the leaderboard on SWE-Bench Lite.

> **Observation 6: Reported results across several studies suggest that integrating search-based inference scaling within agentic frameworks can achieve state-of-the-art performance on some code generation and issue resolution benchmarks.**

# 7 Gaps and Future Directions

Sec. 4 gave an overview of various inference time code reasoning techniques and in Sec. 6 we saw how this impacted code tasks, primarily code generation but also issue resolution. Despite the vast array of work in this domain covered by our survey, there still appear to be gaps which can be explored by the research community. In this section we try to motivate why some of these gaps must be explored and identify a set of Call To Actions (CTA) as specific research directions.

## 7.1 Structure-based CoT understanding and applications

Section 4.1 surveys works that formulate CoT in plan-based, structure-based, and modular methods. As per Sec. 6.1 and Sec. 6.2, structure-based methods outperform plan-based methods and modular methods outperform other Code CoT prompting methods on the code generation benchmarks studied. To further

examine this result, first we must understand why chain-of-thought (CoT) prompting helps over direct prompting. One hypothesis from Prystawski et al. (2023)'s work provides theoretical and experimental evidence that intermediate steps (i.e., chain-of-thought reasoning) reduce bias in transformers. They show that when training data has local structure (as textual data does), intermediate variables (CoT) can outperform direct prediction (no CoT). This suggests that **CoT reasoning helps most when a model is asked to make inferences about concepts that do not co-occur in the training data, but which can be chained together through topics that do.** Section 4.1 surveys works that formulate CoT in plan-based, structure-based, and modular arrangements. The results suggest that structure-aware strategies can outperform plan-based approaches, and modular formats can outperform structure-aware ones. We posit that because code has properties of *structured syntax*, the primitive structures invoked within the CoT are highly local in the training data. Structures (such as idents, branches, loop invariants, functions, etc) are seen countless times in the training corpus. The model's ability to estimate probabilities (and thus its ability to arrive at a correct solution) become sharper by eliciting these localities. Modular structures may push this same principle further. Based on this thesis we present our first call-to-action below.

> **CTA-1\*: Deeper research on code structure and its impact on code tasks will help confirm hypotheses such as: (1) CoT helps by chaining concepts through intermediate topics; (2) code's structured syntax makes primitives (branches, loops, functions) highly local in training data, sharpening probability estimates; (3) modular structures amplify this effect. Validating these may lead to further improvements.**

Code structure improving CoT result should encourage research in exploiting other code properties. Current Code CoT approaches consider a structured representation of the code and also the reasoning and generation of modular solutions. Code CoT approaches should also consider other mechanisms for generating code such as logic programming or event-driven programming (when it applies). Changing the programming paradigm will modify the way the LLM reasons about the solution and might help generate code more aligned to a given specification.

> **CTA-2: Future work should investigate Code CoT approaches that consider other programming paradigms and software design principles for code tasks.**

Inference scaling approaches (Sec. 4.3) sample multiple solutions from LLMs, selecting the best solutions using different strategies. Table 2 shows many approaches use a combination of inference scaling, self-refinement and SWE agents. However there is little to no exploration of inference scaling and Code CoT approaches. The only technique that explores this is PlanSearch which does better than other code CoT techniques on MBPP+ with GPT-4o mini (Fig. 6). Works on inference scaling can also leverage other strategies from CoT such as structure of code and modular code generation that have shown to increase the correctness of the generated solutions, compared to plan-based CoT approaches.

> **CTA-3: Future work should explore combining inference scaling with code-structure based CoT approaches.**

### 7.2 Self-refinement

Self-refinement (Sec. 4.2) involves executing the model generated code and then feeding the execution output back to the model for iterative improvement. This technique outperforms code CoT prompting techniques (Sec. 6.3) on the code generation benchmarks studied. We posit that execution may help because executing code can be used as a deterministic check. Any chain that violates the check can be discarded. Hence, bad chains are filtered out, so variance may collapse faster. However, even with reduced variance, LLMs can still exhibit issues, such as model rigidity. Because code is inherently deterministic (i.e. under certain assumptions, a given input consistently produces the same output), this can lead models to develop rigid generation patterns in training. For example, Twist et al. (2025) show that LLMs exhibit a strong bias towards certain programming languages, like Python; Liu et al. (2025) document the pervasiveness of repeti-

---

\*This CTA stems from hypotheses we formulate based on general trends observed in our analysis, and should be treated as interpretive hypotheses rather than definitive conclusions.

tion in LLM-based code generation, where models often reproduce patterns observed in training. Zhang et al. (2025) demonstrate that LLMs favor certain libraries and APIs by default, reflecting the distribution of their training corpora. Furthermore, Pan et al. (2025b) show that LLMs struggle to generalize to the architectural design principles of given projects, leading to the generation of conflicting code. This phenomenon compels the integration of search in order to explore diverse trajectories, which may explain the recent success of inference scaling techniques.

> **CTA-4⋆: Future work should investigate whether execution serves as a deterministic check that mitigates model rigidity by filtering bad chains and reducing variance. Understanding this mechanism could inform better training data construction and guide the development of more effective search-based inference scaling techniques.**

So far self-refinement approaches have considered execution and unit tests as feedback to self-refine the generated code. To increase the robustness of the generated solutions, these approaches could also reason about other characteristics of the generated code such as efficiency, security, and usability, among others. When generating modularized code, besides reasoning about the functionality of the solution, approaches could also consider reasoning about other important aspects such as maintainability and scalability of the proposed code solution. Mechanisms to validate these properties exist in the generated code can also be used as feedback. Besides unit tests, feedback from integration tests should be considered to further evaluate code modules that were tested as separate units (e.g., with unit testing) work correctly also when interacting with each other.

> **CTA-5: Future work should extend self-refinement approaches beyond unit tests to incorporate software quality metrics (e.g., efficiency, security, maintainability, scalability) and integration testing as feedback mechanisms.**

### 7.3 SWE agents Improvements and Benchmarks

SWE agents (Sec. 4.4) orchestrate LLMs, tools and execution environments resulting in state-of-the-art performance (Sec. 6.5 and Sec. 6.6) on code generation and relatively more challenging issue resolution task. Many approaches covered in our survey also involve multi-agent solutions. Although enabling good results, such agent and multi-agent approaches also increase the degree of complexity of AI systems. Such systems will not be immune to errors, which can propagate through the system and stifle its performance. It is possible that many errors are common across LLMs, agents and tasks. A better understanding and development of techniques to overcome these errors can result in improved performance of multiple systems on multiple tasks.

> **CTA-6: Incorporating a structured way to analyze errors in agents can help them overcome repetitions of the same error, find alternative methods to solve a given step, avoiding waste of resources and increasing their ability to find a solution.**

Overall, patterns of errors made by agents have recently become a topic of growing interest. For instance, Cuadron et al. (2025) examined error patterns in extended internal reasoning chains (i.e., overthinking) in reasoning models, while frameworks such as TRAIL (Deshpande et al., 2025) have focused on categorizing the errors exhibited by agents, and judges the ability of agents to label existing errors in categories. However, these efforts have largely remained at the level of descriptive taxonomies of existing errors. What remains missing is a concrete error benchmark that evaluates agents' ability to recover from their own mistakes or fix injected mistakes. Such a benchmark could be constructed by injecting error instances or categories into a boilerplate agent performing clean tasks, with evaluation metrics including both the raw competence ceiling and the category-wise error recovery rate, alongside latency costs (i.e., the number of steps or seconds required for the agent to recover).

> **CTA-7: Future work should develop error recovery benchmarks that evaluate agents' ability to recover from both self-induced and injected mistakes, measuring category-wise recovery rates and latency costs.**

Beyond agents' errors, another critical frontier is benchmarking tool-use, particularly for SWE agents. While works such as GAIA (Mialon et al., 2023) provide benchmarks for general tool use in generic AI agents, there remains a gap in specific tool-use benchmarks tailored to SWE agents which targets the challenge of system orchestration, specifically. Furthermore, recent work built taxonomies of agentic decision-making pathways (Ceka et al., 2026) showing that agents continue to struggle with certain complexities, performing especially poorly on more advanced SWE-bench problems. They also highlight the presence of "hot nodes"(i.e., critical components in these pathways) that strongly influence whether an agent can successfully generate patches. Based on the aforementioned work, two additional benchmark directions could be: (i) benchmarks that explicitly target complex issues that agents consistently fail on, and (ii) benchmarks, and approaches, that emphasize the specific components of agent decision-making that are most predictive of successful outcomes. For example, TDD-Bench (Otter) (Ahmed et al., 2025) isolates reproduction test case generation as one such critical sub-component of SWE Agents.

> **CTA-8: Despite progress on agents' benchmark development, there remains a gap in specific tool-use benchmarks tailored to SWE agents which targets the challenge of system orchestration; benchmarks that explicitly target complex issues that agents consistently fail on, and benchmarks, and techniques, that emphasize the specific components of agent decision-making that are most predictive of successful outcomes. These are areas where future research should focus on to advance the understanding and performance of Agentic solutions.**

# 8  Limitations

Code reasoning for software engineering is a relatively new and rapidly evolving area. As a result, our survey emphasizes test-time reasoning techniques for which we observed a substantial body of work on code-centric tasks. While we aimed for broad coverage, our taxonomy may not include emerging techniques that have limited adoption or evaluation in the code/SWE literature to date.

Our paper selection process (Sec. 2) relies on keyword- and citation-based search, which cannot guarantee complete recall. Consequently, some relevant papers may be missing. To support transparency and community feedback, we plan to maintain an up-to-date, categorized list of surveyed papers at `https://github.com/AI4Code-IBM-Columbia/code-reasoning-for-swe-tasks`.

Many systems combine multiple reasoning components (e.g., prompting, sampling/search, execution feedback, and refinement). Our taxonomy assigns each paper to a dominant technique based on the primary mechanism emphasized by the method. The primary mechanism can be thought of as core contribution, method or novelty in the publication. To provide a more nuanced view and mitigate potential misrepresentation, we explicitly highlight hybrid approaches in Tab. 2.

This survey focuses on published literature and therefore excludes some highly capable but closed systems. Although such systems may employ code-specific reasoning, their methods are often unclear and their results difficult to verify. In addition, we focus primarily on the effectiveness of different techniques. Cost is an important consideration, especially for test-time compute, but most surveyed papers do not report cost-related metrics. When such metrics are reported, they are not discussed consistently across papers, making comparison difficult.

Finally, due to space constraints and the complexity of many approaches, our descriptions focus on the most salient and broadly applicable ideas rather than all implementation details of every system.

## 8.1  Threats to Validity of Cross-Paper Comparisons

The comparisons synthesized in this survey (Sec. 6) rely on reported results from different papers, which vary in model choice, prompting strategy, inference budgets, benchmark versions (e.g., MBPP vs. MBPP-S vs. MBPP-ET), and evaluation harnesses. These factors introduce confounders that make direct attribution of performance differences to specific techniques difficult. Our comparative analysis does not control for differences in prompts, compute budgets, tool access, or evaluation harnesses, which may affect comparability. To mitigate some of these threats, we condition all comparisons on the same underlying base model and

look for trends that hold consistently across multiple benchmarks, including different splits of the same benchmark (e.g., MBPP, MBPP-S, MBPP-ET) as well as entirely different benchmarks (e.g., HumanEval, APPS). A trend where one technique outperforms another across multiple controlled splits and independent benchmarks is stronger evidence than a single benchmark result. Where trends are supported by many such data points across these controlled same-model comparisons, we present suggestive claims rather than definitive conclusions. Since we try to control for base model across aggregated results, we are limited to the statistics reported by each approach's original paper. Most papers do not report confidence intervals or variance for their results. To help assess the strength of each comparison, we report the margin of difference between approaches where applicable. To facilitate this, we provide a comparison table in the appendix (Tab. 3) that makes it easy to see which specific methods and categories of methods show trends across multiple splits of benchmarks and distinct benchmarks, along with their margins. Future work should aim to verify these trends in a controlled experimental setup with identical conditions across techniques.

## 9 Conclusion

Test or inference-time reasoning techniques have driven major recent gains in AI and have been rapidly adopted in software engineering (SWE). In this survey, we focus on code reasoning for SWE tasks, providing a systematic overview of this emerging area. We begin by proposing a taxonomy of reasoning techniques, including SWE agents (Sec. 4), which defines the scope of code reasoning and clarifies the common design patterns that underlie diverse methods.

Our analysis reveals that many state-of-the-art systems combine multiple reasoning strategies, a trend we summarize in Table 2. Despite this diversity of approaches, we observed that evaluation is concentrated on a small set of code-generation benchmarks, leaving other critical SWE tasks comparatively underexplored. To address this imbalance, we catalog a broader set of tasks and benchmarks for code reasoning (Sec. 5).

To assess the impact of existing techniques, we perform a comparative analysis over commonly used models and benchmarks (Sec. 6), highlighting relative strengths of different code reasoning methods. These observations motivate several future research directions, including the design of more realistic and comprehensive SWE benchmarks and exploration of reasoning and agent architectures that better capture the constraints and real-world engineering workflows, while taking advantage of features that formal languages afford (Sec. 7). We hope this survey serves as a foundation for the growing field of code reasoning for SWE and supports the community in building more capable, reliable, and practical AI tools.

### Acknowledgments

This work was supported in part by the National Science Foundation Graduate Research Fellowship Program (NSF GRFP), IBM, NSF CNS-2247370, NSF CCF-2313055, DARPA/NIWC Pacific N66001-21-C-4018. Any opinions, findings, conclusions or recommendations expressed herein are those of the authors and do not necessarily reflect those of the US Government, NSF, IBM, or DARPA.

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

# A  Appendix

## A.1  Paper Roadmap

## A.2   Benchmarks

*HumanEval (HE)* Chen et al. (2021a) is a set of 164 hand-written programming problems. Each problem includes a function signature, docstring, body, and several unit tests, with an average of 7.7 tests per problem. A multi-language version of HE is also available in HumanEval-XL Peng et al. (2024).

*MBPP* Austin et al. (2021) (The Most Basic Programming Problems) benchmark has 1k crowd-sourced Python programming problems and was designed to be solvable by entry level programmers. Each problem consists of a task description, code solution and three automated test cases. *EvalPlus* Liu et al. (2023) augments a given evaluation dataset with large amounts of new test cases created by an automatic test input generator, powered by both LLM- and mutation-based strategies. EvalPlus includes *MBPP+, HumanEval+*, and *EvalPerf*.

*APPS* Hendrycks et al. (2021b) is another benchmark for code generation with 10k samples that measures the ability of models to take an arbitrary natural language specification and generate satisfactory Python code. More recent extensions of some of the above benchmarks such as *HumanEval-ET, MBPP-ET*, and *APPS-ET* were introduced by Dong et al. (2025), where the amount of correct test cases were extended for each benchmark 100+ on average according to the reference code.

*CodeContests* Li et al. (2022b) is a code generation dataset with problems curated from competitive programming platforms such as Codeforces, requiring solutions to challenging code generation problems. This dataset has solutions to the given problems in Python, Java, and C++, with an English description of the code problems.

## A.3   Code Evaluation

To address the poor correlation with human evaluation of exact or fuzzy match metrics, ICE-Score was recently proposed as an evaluation metric that instructs LLMs for code assessments Zhuo (2024). The ICE-Score evaluation showed superior correlations with functional correctness and human preferences, without the need for test oracles or references. The efficacy of ICE-Score was measured w.r.t. human preference and execution success for four programming languages.

Additionally, CodeScore Dong et al. (2025) is another code evaluation metric that was recently proposed to measure the functional correctness of generated codes on three input formats (Ref-only, NL-only, and

Ref&NL). CodeScore can be obtained through the UniCE framework that assists models in learning code execution and predicting an estimate of execution PassRatio.

## A.4 Metrics

Functional correctness of generated code by LLMs is mainly measured by passing tests. One of the basic metrics to measure the correctness of code is the percentage of tasks in a given benchmark where the generated code successfully passes all tests. Chen et al. (2021a) shows that exact or fuzzy match metrics (e.g., BLEU) are not adequate or reliable indicators of functional correctness of code, by showing that functionally different programs generated by a model often have higher BLEU scores than functionally equivalent ones.

The metric *pass@k* is the probability of generating at least one solution passing all test cases successfully in $k$ trials. The *AvgPassRatio* measures the degree of correctness of generated code on evaluation test cases, it considers whether the generated code is completely correct on evaluation test cases or not. Another metric is the percentage of problems solved using $n$ submissions from $k$ samples per problem, denoted as $n@k$.

## A.5 Comparison Margins

| Method A (Category) | Method B (Category) | Winner (Category) | Bench. | Model | Margin |
|---|---|---|---|---|---|
| **Observation 1: Structure CoT vs. Plan CoT** | | | | | |
| CGO (Struct. CoT) | Self-Planning (Plan CoT) | CGO (Struct. CoT) | MBPP-S | gpt-3.5-turbo | +3.4 |
| CGO (Struct. CoT) | Self-Planning (Plan CoT) | CGO (Struct. CoT) | MBPP+ | gpt-3.5-turbo | +5.6 |
| CGO (Struct. CoT) | Self-Planning (Plan CoT) | CGO (Struct. CoT) | HE | gpt-3.5-turbo | +1.9 |
| CGO (Struct. CoT) | Self-Planning (Plan CoT) | CGO (Struct. CoT) | HE+ | gpt-3.5-turbo | +1.2 |
| CGO (Struct. CoT) | ClarifyGPT (Plan CoT) | CGO (Struct. CoT) | MBPP-S | gpt-3.5-turbo | +11.9 |
| CGO (Struct. CoT) | ClarifyGPT (Plan CoT) | CGO (Struct. CoT) | HE | gpt-3.5-turbo | +0.2 |
| CGO (Struct. CoT) | Self-Planning (Plan CoT) | CGO (Struct. CoT) | MBPP-S | Llama-3-8B | +0.2 |
| CGO (Struct. CoT) | Self-Planning (Plan CoT) | CGO (Struct. CoT) | MBPP+ | Llama-3-8B | +1.0 |
| SCoT (Struct. CoT) | Self-Planning (Plan CoT) | SCoT (Struct. CoT) | MBPP | gpt-4o-mini | +11.8 |
| SCoT (Struct. CoT) | Self-Planning (Plan CoT) | SCoT (Struct. CoT) | MBPP+ | gpt-4o-mini | +9.0 |
| SCoT (Struct. CoT) | Self-Planning (Plan CoT) | SCoT (Struct. CoT) | APPS | gpt-3.5-turbo | +0.7 |
| **Observation 2: Modular CoT vs. Structure/Plan CoT** | | | | | |
| MoT (Mod. CoT) | SCoT (Struct. CoT) | MoT (Mod. CoT) | MBPP | DeepSeek-R1 | +17.0 |
| MoT (Mod. CoT) | Self-Planning (Plan CoT) | MoT (Mod. CoT) | MBPP | DeepSeek-R1 | +6.5 |
| MoT (Mod. CoT) | SCoT (Struct. CoT) | MoT (Mod. CoT) | HE | DeepSeek-R1 | +10.3 |
| MoT (Mod. CoT) | Self-Planning (Plan CoT) | MoT (Mod. CoT) | HE | DeepSeek-R1 | +9.7 |
| MoT (Mod. CoT) | SCoT (Struct. CoT) | MoT (Mod. CoT) | MBPP | gpt-4o-mini | +10.0 |
| MoT (Mod. CoT) | SCoT (Struct. CoT) | MoT (Mod. CoT) | MBPP+ | gpt-4o-mini | +6.7 |
| CodeChain (Mod. CoT) | SCoT (Struct. CoT) | CodeChain (Mod. CoT) | APPS | gpt-3.5-turbo | +4.4 |
| CodeChain (Mod. CoT) | Self-Planning (Plan CoT) | CodeChain (Mod. CoT) | APPS | gpt-3.5-turbo | +5.1 |
| **Observation 3: Self-refinement vs. Code CoT** | | | | | |
| $\mu$-Fix (Self-refine.) | CGO (Struct. CoT) | $\mu$-Fix (Self-refine.) | HE | gpt-3.5-turbo | +15.6 |
| $\mu$-Fix (Self-refine.) | SCoT (Struct. CoT) | $\mu$-Fix (Self-refine.) | HE | gpt-3.5-turbo | +29.6 |
| $\mu$-Fix (Self-refine.) | Self-Planning (Plan CoT) | $\mu$-Fix (Self-refine.) | HE | gpt-3.5-turbo | +17.5 |
| $\mu$-Fix (Self-refine.) | ClarifyGPT (Plan CoT) | $\mu$-Fix (Self-refine.) | HE | gpt-3.5-turbo | +15.8 |
| Self-Debug (Self-refine.) | ClarifyGPT (Plan CoT) | Self-Debug (Self-refine.) | MBPP-ET | gpt-3.5-turbo | +4.8 |
| Self-Debug (Self-refine.) | SCoT (Struct. CoT) | Self-Debug (Self-refine.) | MBPP | gpt-3.5-turbo | +27.2 |
| Rev. Self-Debug (Self-refine.) | PlanSearch (Code CoT) | Rev. Self-Debug (Self-refine.) | HE+ | Claude-3.5 | +7.4 |
| $\mu$-Fix (Self-refine.) | CodeChain (Mod. CoT) | $\mu$-Fix (Self-refine.) | APPS | gpt-3.5-turbo | +9.3 |
| $\mu$-Fix (Self-refine.) | SCoT (Struct. CoT) | $\mu$-Fix (Self-refine.) | APPS | gpt-3.5-turbo | +13.7 |
| $\mu$-Fix (Self-refine.) | Self-Planning (Plan CoT) | $\mu$-Fix (Self-refine.) | APPS | gpt-3.5-turbo | +14.4 |
| **Observation 4: Inference Scaling vs. Code CoT** | | | | | |
| ORPS (Inf. Scaling) | MoT (Mod. CoT) | ORPS (Inf. Scaling) | MBPP | gpt-4o-mini | +21.8 |

| Method A (Category) | Method B (Category) | Winner (Category) | Bench. | Model | Margin |
|---|---|---|---|---|---|
| ORPS (Inf. Scaling) | SCoT (Struct. CoT) | ORPS (Inf. Scaling) | MBPP | gpt-4o-mini | +31.8 |
| ORPS (Inf. Scaling) | Self-Planning (Plan CoT) | ORPS (Inf. Scaling) | MBPP | gpt-4o-mini | +43.6 |
| REx (Inf. Scaling) | CodeChain (Mod. CoT) | REx (Inf. Scaling) | APPS | gpt-4 | +8.5 |
| S* (Inf. Scaling) | PlanSearch (Code CoT) | S* (Inf. Scaling) | LCB | o1-mini | +15.8 |
| S* (Inf. Scaling) | PlanSearch (Code CoT) | S* (Inf. Scaling) | LCB | gpt-4o-mini | +22.3 |
| PlanSearch (Code CoT♣) | MoT (Mod. CoT) | PlanSearch (Code CoT♣) | MBPP+ | gpt-4o-mini | +15.4 |
| PlanSearch (Code CoT♣) | SCoT (Struct. CoT) | PlanSearch (Code CoT♣) | MBPP+ | gpt-4o-mini | +22.1 |
| PlanSearch (Code CoT♣) | Self-Planning (Plan CoT) | PlanSearch (Code CoT♣) | MBPP+ | gpt-4o-mini | +31.1 |
| CodeTree (Inf. Scaling) | MoT (Mod. CoT) | CodeTree (Inf. Scaling) | MBPP+ | gpt-4o-mini | +18.9 |
| CodeTree (Inf. Scaling) | SCoT (Struct. CoT) | CodeTree (Inf. Scaling) | MBPP+ | gpt-4o-mini | +25.6 |
| CodeTree (Inf. Scaling) | Self-Planning (Plan CoT) | CodeTree (Inf. Scaling) | MBPP+ | gpt-4o-mini | +34.6 |
| **Observation 5: SWE Agents vs. other approaches** | | | | | |
| PairCoder (SWE Agent) | ClarifyGPT (Plan CoT) | PairCoder (SWE Agent) | HE | gpt-4 | +6.1 |
| AgileCoder (SWE Agent) | ClarifyGPT (Plan CoT) | AgileCoder (SWE Agent) | HE | gpt-4 | +3.1 |
| PairCoder (SWE Agent) | ClarifyGPT (Plan CoT) | PairCoder (SWE Agent) | MBPP-S | gpt-4 | +12.5 |
| PairCoder (SWE Agent) | CGO (Struct. CoT) | PairCoder (SWE Agent) | MBPP+ | gpt-3.5-turbo | +4.0 |
| PairCoder (SWE Agent) | Self-Planning (Plan CoT) | PairCoder (SWE Agent) | MBPP+ | gpt-3.5-turbo | +9.6 |
| PairCoder (SWE Agent) | SCoT (Struct. CoT) | PairCoder (SWE Agent) | MBPP | gpt-3.5-turbo | +33.6 |
| AgileCoder (SWE Agent) | SCoT (Struct. CoT) | AgileCoder (SWE Agent) | MBPP | gpt-3.5-turbo | +33.9 |
| PairCoder (SWE Agent) | Self-Debug (Self-refine.) | PairCoder (SWE Agent) | MBPP | gpt-3.5-turbo | +6.4 |
| AgileCoder (SWE Agent) | Self-Debug (Self-refine.) | AgileCoder (SWE Agent) | MBPP | gpt-3.5-turbo | +6.7 |
| PairCoder (SWE Agent) | $\mu$-Fix (Self-refine.) | PairCoder (SWE Agent) | HE | DeepSeek-Coder | +1.9 |
| PairCoder (SWE Agent) | Self-Debug (Self-refine.) | PairCoder (SWE Agent) | HE | DeepSeek-Coder | +8.0 |
| PairCoder (SWE Agent) | UniCoder (Struct. CoT) | PairCoder (SWE Agent) | HE | DeepSeek-Coder | +14.8 |
| **Observation 6: Agents + Inference Scaling vs. other approaches** | | | | | |
| CodeTree (Agent+Inf. Sc.) | MoT (Mod. CoT) | CodeTree (Agent+Inf. Sc.) | HE+ | gpt-4o-mini | +1.3 |
| CodeTree (Agent+Inf. Sc.) | SCoT (Struct. CoT) | CodeTree (Agent+Inf. Sc.) | HE+ | gpt-4o-mini | +6.1 |
| CodeTree (Agent+Inf. Sc.) | Self-Planning (Plan CoT) | CodeTree (Agent+Inf. Sc.) | HE+ | gpt-4o-mini | +4.9 |
| CodeTree (Agent+Inf. Sc.) | PlanSearch (Code CoT) | CodeTree (Agent+Inf. Sc.) | HE+ | gpt-4o-mini | +1.1 |
| CodeTree (Agent+Inf. Sc.) | MoT (Mod. CoT) | CodeTree (Agent+Inf. Sc.) | MBPP+ | gpt-4o-mini | +18.9 |
| CodeTree (Agent+Inf. Sc.) | SCoT (Struct. CoT) | CodeTree (Agent+Inf. Sc.) | MBPP+ | gpt-4o-mini | +25.6 |
| CodeTree (Agent+Inf. Sc.) | Self-Planning (Plan CoT) | CodeTree (Agent+Inf. Sc.) | MBPP+ | gpt-4o-mini | +34.6 |
| CodeTree (Agent+Inf. Sc.) | PlanSearch (Code CoT) | CodeTree (Agent+Inf. Sc.) | MBPP+ | gpt-4o-mini | +3.5 |
| SWE-Search (Agent+Inf. Sc.) | MASAI (SWE Agent) | SWE-Search (Agent+Inf. Sc.) | SWE-Lite | gpt-4o | +2.7 |
| SWE-Search (Agent+Inf. Sc.) | Agentless (SWE Agent) | SWE-Search (Agent+Inf. Sc.) | SWE-Lite | gpt-4o | +6.7 |
| SWE-Search (Agent+Inf. Sc.) | AutoCodeRover (SWE Agent) | SWE-Search (Agent+Inf. Sc.) | SWE-Lite | gpt-4o | +8.3 |
| SWE-Search (Agent+Inf. Sc.) | OpenHands (SWE Agent) | SWE-Search (Agent+Inf. Sc.) | SWE-Lite | gpt-4o | +9.0 |
| SWE-Search (Agent+Inf. Sc.) | SWE-Agent (SWE Agent) | SWE-Search (Agent+Inf. Sc.) | SWE-Lite | gpt-4o | +12.7 |

Table 3: Margins of difference for comparisons discussed in Sec. 6. Each row shows the two approaches being compared (with their technique category), the winner, the benchmark, the model, and the margin (in percentage points). Metrics are pass@1 for code generation benchmarks and resolved rate for SWE-Bench. All results are sourced from each approach's original paper.

♣ PlanSearch is categorized as Code CoT (Tab. 2) but is a notable outlier among CoT methods, exhibiting large margins over other CoT approaches; given these margins, the role of inference scaling as a secondary component in PlanSearch warrants consideration.

## A.6 Survey Coverage Justifications

| Survey | Dimension | Justification |
|---|---|---|
| Dong et al., 2022 | Reasoning ✓ | References CoT reasoning but does not expand on it; reasoning is not the paper's focus. |
| | All others ✗ | Surveys in-context learning broadly; does not cover SWE tasks, agents, taxonomy, or benchmarks for code. |
| Qiao et al., 2022 | Reasoning ✓ | Extensively studies CoT and other prompting approaches with a detailed taxonomy of reasoning methods. |
| | All others ✗ | Does not include software engineering tasks; taxonomies and benchmarks are for general reasoning (e.g., arithmetic, commonsense), not code. |
| Huang & Chang, 2022 | Reasoning ✓ | Extensively studies CoT and other prompting approaches covering deductive, inductive, and analogical reasoning. |
| | All others ✗ | Does not include software engineering tasks; benchmarks are for general reasoning, not code. |
| Zan et al., 2022 | SWE Tasks ✓ | Surveys 27 LLMs for natural language to code generation; covers code generation only, which is partial SWE coverage. |
| | All others ✗ | Does not discuss reasoning techniques. Catalogs code generation benchmarks but not in the context of evaluating reasoning approaches. |
| Chu et al., 2023 | Reasoning ✓ | Covers CoT extensively and defines XoT (Tree-of-Thought, Program-of-Thought, etc.), but limited to the CoT family; does not cover execution-based refinement, inference scaling, or agentic reasoning. |
| | SWE Tasks ✓ | Mentions Program-of-Thought and code generation, but focuses on reasoning benchmarks rather than SWE benchmarks. |
| Jiang et al., 2024a | Reasoning ✓ | Extensive code generation survey covering many research topics; touches reasoning but it is not the primary focus. |
| | SWE Tasks ✓ | Covers code generation extensively but not broader SWE tasks (e.g., issue resolution, test generation). |
| Sun et al. (2024a) | Reasoning ✓ | Wide-ranging survey covering 50 models with comprehensive coverage of techniques. |
| | SWE Tasks ✓ | Covers 20 different code-related task categories. |
| | Benchmarks ✓ | Covers benchmarks but not with the depth of a dedicated benchmark survey. |
| Plaat et al., 2024 | Reasoning ✓ | Classifies in-context reasoning into prompting, evaluating, and control (inference scaling and search) based strategies. |
| | All others ✗ | Does not focus on coding tasks. Any code or agent mentions are incidental to general reasoning. Taxonomy is for general reasoning steps, not code-specific reasoning techniques. |
| Wang et al. (2024b) | SWE Tasks ✓ | Focuses exclusively on reinforcement learning in code generation; covers code generation only. |
| | Benchmarks ✓ | Covers code generation benchmarks within the RL context. |
| Chen et al. (2024b) | SWE Tasks ✓ | Comprehensive coverage of evaluation techniques across coding tasks. |
| | Benchmarks ✓ | Covers benchmarks as part of evaluation, but evaluation methodology is the focus. |
| Yehudai et al. (2025) | SWE Tasks ✓ | Comprehensive coverage of SWE tasks through agent evaluation. |
| | Agents ✓ | Dedicated focus on LLM-agents including SWE Agents. |
| | Benchmarks ✓ | Comprehensive benchmark coverage for agent evaluation. |
| Xu et al., 2025 | Reasoning ✓ | Discusses reinforcement learning based reasoning techniques comprehensively. |
| | All others ✗ | Does not discuss code-specific reasoning strategies, SWE tasks, agents, or code benchmarks. |
| Huynh & Lin, 2025 | Reasoning ✓ | Surveys many topics in the space; touches reasoning but not deeply. |
| | SWE Tasks ✓ | Covers SWE topics including challenges and applications, but not comprehensively. |

| Survey | Dimension | Justification |
|---|---|---|
| | Reasoning ✓ | Covers reasoning through code-assisted methods (e.g., PoT), but lacks full coverage of broader reasoning paradigms (e.g., RL, inference scaling). |
| Yang et al., 2025 | SWE Tasks ✓ | Includes code generation, debugging, and agents as part of the code-reasoning interplay, but SWE tasks are not the primary organizational lens. |
| | Agents ✓ | Discusses agents in the context of code-reasoning interaction, but no dedicated or in-depth agent taxonomy. |
| | Taxonomy ✓ | Provides a taxonomy of code-reasoning interplay, not a general taxonomy of reasoning or SWE systems. |
| | Benchmarks ✓ | Covers both code and reasoning benchmarks, but without deep or systematic evaluation. |
| | Reasoning ✅ | Comprehensive survey on context engineering that covers reasoning within its framework. |
| Mei et al., 2025 | SWE Tasks ✓ | Covers many aspects of SWE reasoning and tasks, but has a much more general focus and is unable to cover many aspects of code-specific reasoning. |
| | Agents ✓ | Covers agents within context engineering, but not with SWE-agent specific depth. |
| | Taxonomy ✓ | Provides a taxonomy for context engineering, not for code reasoning specifically. |
| | Benchmarks ✓ | Mentions SWE-Bench and other benchmarks but without detailed analysis. |

Table 4: Justifications for coverage ratings in Tab. 1. Each entry explains why a survey received its rating (❌ = not covered, ✓ = partial, ✅ = comprehensive) for a given dimension, evaluated specifically for code and SWE tasks.

### A.7    Results Tables

| Approach | Model | APPS Introductory | APPS Interview | APPS Competition | APPS-ET | APPS |
|---|---|---|---|---|---|---|
| CodeChain Le et al. (2023) | gpt-4 | 71.1 | 55.0 | 23.3 | – | 61.5 |
| | gpt-3.5-turbo-16k | 54.5 | 28.1 | 12.4 | – | 26.4 |
| | WizardCoder | 26.3 | 7.5 | 3.8 | – | 10.5 |
| ChainCoder ◊ Zheng et al. (2023) | ChainCoder-1B | 17.5 | 7.4 | 5.5 | – | – |
| AlphaCode ◊ Li et al. (2022a) | AlphaCode-1B | 14.4 | 5.6 | 4.6 | – | – |
| Self-Planning Jiang et al. (2024b) | gpt-3.5-turbo | – | – | – | 8.3 | 21.3 |
| | DeepSeekCoder | – | – | – | 1.0 | 4.0 |
| SCoT (Li et al., 2025b) | gpt-3.5-turbo | – | – | – | 7.7 | 22.0 |
| | DeepSeek-Coder-6.7B-Instr | – | – | – | 1.3 | 4.3 |
| Self-Debugging (Chen et al., 2024c) | gpt-3.5-turbo | – | – | – | 6.2 | 18.7 |
| | DeepSeek-Coder-6.7B-Instr | – | – | – | 1.3 | 4.7 |
| CYCLE (Ding et al., 2024a) | CYCLE-350M | – | – | – | – | 8.7 |
| | CYCLE-1B | – | – | – | – | 10.9 |
| | CYCLE-2.7B | – | – | – | – | 11.6 |
| | CYCLE-3B | – | – | – | – | 11.3 |
| $\mu$-Fix (Tian et al., 2025) | gpt-3.5-turbo | – | – | – | 10.3 | 35.7 |
| | DeepSeek-Coder-6.7B-Instr | – | – | – | 5.0 | 14.0 |
| REx Tang et al. (2024) | gpt-4 | – | – | – | – | $\sim 70$ |

Table 5: Performance across the APPS benchmark Hendrycks et al. (2021a), including the **APPS Introductory**, **Interview**, **Competition**, **APPS-ET**, and **APPS** overall sets. Default performance is reported as *pass*@1 (%). Approaches marked with ◊ use the $n@k$ metric, where $n = 5$ and $k = 1,000$.

| Approach | Model | LCB | CodeContests | M³ToolEval |
|---|---|---|---|---|
| S* (Li et al., 2025a) | Qwen-2.5-Coder-Instruct 32B | 70.1 | 21.8 | - |
| | gpt-4o-mini | 61.3 | 23.0 | - |
| | R1-Distill-32B | 85.7 | - | - |
| | o1-mini | 85.3 | 48.5 | - |
| PlanSearch Wang et al. (2024a) | DeepSeek-Coder-V2 | 41.4 | – | - |
| | gpt-4o-mini | 39.0 | – | - |
| | gpt-4o | 41.3 | – | - |
| | Claude-Sonnet-3.5 | 40.3 | - | - |
| | o1-mini | 69.5 | – | - |
| CodeChain † Le et al. (2023) | gpt-3.5 | - | 14.1 | - |
| ChainCoder ‡ Zheng et al. (2023) | ChainCoder-1B | - | $\sim 15$ | - |
| AlphaCode ‡ Li et al. (2022a) | AlphaCode-9B | - | 14.3 | - |
| | AlphaCode-41B | - | 15.6 | - |
| PairCoder Zhang et al. (2024a) | gpt-3.5-turbo | - | 15.2 | - |
| | DeepSeek-Coder | - | 14.6 | - |
| CodeTree Zhang et al. (2024a) | gpt-4o-mini | - | 26.4 | - |
| | gpt-4o | - | 43.0 | - |
| | Llama-3.1-8B | - | 12.1 | - |
| AlphaCodium † Ridnik et al. (2024) | DeepSeek-33B | - | 24.0 | - |
| | gpt-3.5 | - | 17.0 | - |
| | gpt-4 | - | 29.0 | - |
| CodeAct Wang et al. (2024c) | gpt-4 | – | – | 74.4 |
| Tree-of-Code Ni et al. (2024) | Mix-modal | – | – | 81.6 |

Table 6: Performance across the **LiveCodeBench (LCB)**, **CodeContests (test set)**, and **M³ToolEval**. Default results are reported as *pass*@1. Approaches marked with † indicate *pass*@5, while those marked with ‡ use the $n@k$ of 10@1$k$ rate. S* results reflect performance on LCB v2.

| Approach | Model | SWE-Bench Verified | SWE-Bench Lite | SWE-Bench |
|---|---|---|---|---|
| Agentless Xia et al. (2024) | gpt-4o | 33.2 | 24.3 | - |
| | o1-preview | 41.3 | - | - |
| | DeepSeek-V3 | 42.0 | - | - |
| | DeepSeek-R1 | 49.2 | - | - |
| | Claude-3.5-Sonnet | 53.0 | - | - |
| AutoCodeRover Zhang et al. (2024b) | Qwen2-72B-Instruct | - | 9.3 | - |
| | gpt-4o | 28.8 | 22.7 | - |
| | gpt-4 | - | 19.0 | - |
| MASAI Arora et al. (2024) | gpt-4o | – | 28.3 | - |
| SWE-Agent Yang et al. (2024b) | Claude-3.5-Sonnet | 33.6 | 23.0 | - |
| | gpt-4o | 23.2 | 18.3 | - |
| SWE-Gym Pan et al. (2024) | Qwen-2.5-Coder-Instruct 32B | 20.6 | 15.3 | - |
| | SWE-Gym-32B | 32.0 | 26.0 | - |
| SWE-Search Antoniades et al. (2024) | gpt-4o | - | 31.0 | - |
| | gpt-4o-mini | - | 17.0 | - |
| | Qwen-2.5-72b-Instruct | - | 24.7 | - |
| | Deepseek-V2.5 | - | 21.0 | - |
| | Llama-3.1-70b-Instruct | - | 17.7 | - |
| Lingma Ma et al. (2024) | Lingma SWE-GPT 72B | 30.2 | 22.0 | - |
| | Lingma SWE-GPT 7B | 18.2 | 12.0 | - |
| SWE-Fixer Xie et al. (2025) | SWE-Fixer-72B | 32.8 | 24.7 | - |
| HyperAgent Phan et al. (2024) | - | 33.0 | 26.0 | - |
| SWE-RL Wei et al. (2025) | Llama3-SWE-RL-70B | 41.0 | - | - |
| CodeR Chen et al. (2024a) | gpt-4 | – | 28.3 | - |
| CodeTree Li et al. (2024) | gpt-4o-mini | – | – | 27.6 |
| OpenHands Wang et al. (2025b) | gpt-4o-mini | – | 7.0 | - |
| | gpt-4o | – | 22.0 | - |
| | Claude-3.5-Sonnet | – | 26.0 | - |

Table 7: Performance on **SWE-Bench Verified**, and **SWE-Bench Lite**, and **SWE-Bench**. Performance is measured by resolved rate.

| Approach | Model | MBPP+ | MBPP | MBPP-ET | MBPP-S |
|---|---|---|---|---|---|
| PlanSearch Wang et al. (2024a) | gpt-4o-mini | 73.5 | – | – | – |
| | gpt-4o | 77.2 | – | – | – |
| | DeepSeekCoder-V2 | 76.3 | – | – | – |
| | Claude-3.5-sonnet | 77.1 | – | – | – |
| ClarifyGPT Mu et al. (2023) | gpt-3.5-turbo | – | – | 55.6 | 74.1 |
| | gpt-4 | – | – | 58.5 | 78.7 |
| Self-Planning Jiang et al. (2024b) | Codex | – | – | 41.9 | 55.7 |
| | gpt-4o-mini | 42.4 | 52.1 | 48.2 | – |
| | DeepSeek-R1 | 55.4 | 68.4 | 65.5 | – |
| | gpt-3.5-turbo | 68.1 | – | – | 82.6 |
| | Llama-3 8B Instr. | 56.9 | – | – | 67.9 |
| SCoT (Li et al., 2025b) | gpt-3.5-turbo | – | 47.0 | – | – |
| | Codex | – | 38.3 | – | – |
| | gpt-4o-mini | 51.4 | 63.9 | 55.6 | – |
| | DeepSeek-R1 | 46.9 | 57.9 | 61.3 | – |
| MoT Pan & Zhang (2025) | DeepSeek-R1 | 60.4 | 74.9 | 68.0 | – |
| | gpt-4o-mini | 58.1 | 73.9 | 58.9 | – |
| CGO Yeo et al. (2025) | gpt-3.5-turbo | 73.7 | – | – | 86.0 |
| | Llama-3 8B Instr. | 57.9 | – | – | 68.1 |
| UniCoder Sun et al. (2024b) | Deepseek-Coder | – | 64.3 | – | – |
| | CodeLlama-7B | – | 65.2 | – | – |
| Self-Debugging Chen et al. (2024c) | Codex | – | 70.8 | – | – |
| | gpt-3.5-turbo | – | 74.2 | 60.4 | – |
| | gpt-4 | – | 80.6 | – | – |
| | StarCoder | – | 53.2 | – | – |
| | DeepSeek-Coder-6.7B-Instruct | – | – | 56.9 | – |
| LeDex Jiang et al. (2025) | StarCoder-15B | 54.3 | 58.2 | – | – |
| | CodeLlama-7B | 52.9 | 58.1 | – | – |
| | CodeLlama-13B | 57.9 | 61.9 | – | – |
| Revisiting Self-Debugging Chen et al. (2025b) | gpt-4o | 76.5 | 91.5 | – | – |
| | Claude-3.5-sonnet | 77.0 | 92.6 | – | – |
| | Llama-3-70B-Instr. | 71.2 | 84.4 | – | – |
| | Qwen-2.5-Coder-7B-Instr | 70.6 | 84.7 | – | – |
| ORPS Yu et al. (2024) | Llama-3.1-8B-Instruct | - | 90.4 | – | – |
| | DeepSeek-Coder-7B-Instruct-v1.5 | - | 93.0 | – | – |
| | Qwen-2.5-Coder-7B-Instruct | - | 94.9 | – | – |
| | Qwen-2.5-Coder-14B-Instruct | - | 95.3 | – | – |
| | gpt-4o-mini | - | 95.7 | – | – |

Table 8: Performance on the **MBPP +**, **MBPP**, **MBPP-ET**, and **MBPP-sanitized** benchmarks. All results are reported as *pass*@1.

| Approach | Model | MBPP+ | MBPP | MBPP-ET | MBPP-S |
|---|---|---|---|---|---|
| CodeTree Li et al. (2024) | gpt-4o-mini | 77.0 | 96.8 | – | – |
| | gpt-4o | 80.7 | 98.7 | – | – |
| | Llama-3.1-8B-Instr. | 73.3 | 90.5 | – | – |
| AgileCoder (Nguyen et al., 2024) | gpt-3.5-turbo | – | 80.9 | – | – |
| | claude-3-haiku | – | 84.3 | – | – |
| PairCoder (Zhang et al., 2024a) | gpt-3.5-turbo | 77.7 | 80.6 | – | – |
| | DeepSeek-Coder | 75.7 | 78.8 | – | – |
| | gpt-4 | – | – | – | 91.2 |
| CYCLE (Ding et al., 2024a) | CYCLE-350M | – | – | – | 32.6 |
| | CYCLE-1B | – | – | – | 35.8 |
| | CYCLE-2.7B | – | – | – | 48.5 |
| | CYCLE-3B | – | – | – | 51.3 |
| $\mu$-Fix (Tian et al., 2025) | gpt-3.5-turbo | – | – | 69.1 | – |
| | DeepSeek-Coder-6.7B-Instruct | – | – | 63.3 | – |
| SemCoder (Ding et al., 2024b) | SemCoder-S-6.7B | 68.5 | 79.6 | – | – |
| | SemCoder-6.7B | 65.3 | 79.9 | – | – |

Table 9: Performance on the **MBPP +**, **MBPP**, **MBPP-ET**, and **MBPP-sanitized** benchmarks (continued). All results are reported as $pass@1$.

| Approach | Model | HE+ | HE | HE-XL | HE-X | HE-ET |
|---|---|---|---|---|---|---|
| PlanSearch Wang et al. (2024a) | gpt-4o-mini | 83.7 | – | – | – | – |
| | gpt-4o | 86.4 | – | – | – | – |
| | DeepSeekCoder-V2 | 82.8 | – | – | – | – |
| | Claude-3.5-sonnet | 81.6 | – | – | – | – |
| ClarifyGPT Mu et al. (2023) | gpt-3.5-turbo | – | 74.4 | – | – | 64.8 |
| | gpt-4 | – | 87.8 | – | – | 78.1 |
| Self-Planning Jiang et al. (2024b) | Codex | – | 60.3 | – | 60.3 | 46.2 |
| | gpt-4o-mini | 79.9 | 87.2 | – | – | 87.1 |
| | DeepSeek-R1 | 79.3 | 85.4 | – | – | 85.3 |
| | gpt-3.5-turbo | 67.3 | 72.7 | – | – | – |
| | LLaMA-3 8B Instr. | 52.8 | 60.1 | – | – | – |
| SCoT (Li et al., 2025b) | gpt-3.5-turbo | – | 60.6 | – | – | – |
| | Codex | – | 49.8 | – | – | – |
| | gpt-4o-mini | 78.7 | 86.6 | – | – | 86.0 |
| | DeepSeek-R1 | 79.3 | 84.8 | – | – | – |
| | DeepSeekCoder | – | – | 69.3 | – | – |
| | Qwen-2.5-Coder | – | – | 74.4 | – | – |
| MoT Pan & Zhang (2025) | DeepSeek-R1 | 88.4 | 95.1 | – | – | 94.5 |
| | gpt-4o-mini | 83.5 | 92.1 | – | – | 91.5 |
| MSCoT Pan & Zhang (2025) | DeepSeek-Coder | – | – | 66.0 | – | – |
| | Qwen2.5-Coder | – | – | 72.3 | – | – |
| CGO Yeo et al. (2025) | gpt-3.5-turbo | 68.5 | 74.6 | – | – | – |
| | LLaMA-3 8B Instr. | 56.2 | 62.4 | – | – | – |
| UniCoder Sun et al. (2024b) | DeepSeek-Coder | – | 70.6 | – | – | – |
| | CodeLlama-7B | – | 65.4 | – | – | – |
| COTTON Yang et al. (2024a) | gpt-3.5-turbo | 76.2 | 74.4 | – | – | – |
| | DeepSeekCoder | – | – | 61.8 | – | – |
| | Qwen-2.5-Coder | – | – | 68.7 | – | – |

Table 10: Performance on the **HumanEval+**, **HumanEval**, **HumanEval-XL**, **HumanEval-X**, and **HumanEval-ET** benchmarks. All results are reported as *pass*@1.

| Approach | Model | HE+ | HE | HE-XL | HE-X | HE-ET |
|----------|-------|-----|-----|-------|------|-------|
| Agile Coder Nguyen et al. (2024) | gpt-3.5-turbo | − | 70.5 | − | − | − |
| | claude-3-haiku | − | 79.3 | − | − | − |
| | gpt-4 | − | 90.9 | − | − | − |
| CodeAct Wang et al. (2024c) | CodeActAgent(LLaMA-2-7B) | − | 18.1 | − | − | − |
| | CodeActAgent(Mistral-7B) | − | 34.7 | − | − | − |
| PairCoder Zhang et al. (2024a) | gpt-3.5-turbo | 77.4 | 87.8 | − | − | − |
| | DeepSeek-Coder | 76.2 | 85.4 | − | − | − |
| | gpt-4 | − | 93.9 | − | − | − |
| CodeTree Li et al. (2024) | gpt-4o-mini | 84.8 | 94.5 | − | − | − |
| | gpt-4o | 86.0 | 94.5 | − | − | − |
| | Llama-3.1-8B | 72.0 | 82.3 | − | − | − |
| $\mu$-Fix Tian et al. (2025) | gpt-3.5-turbo | 80.5 | 90.2 | − | − | 79.9 |
| | DeepSeek-Coder-6.7B-Instr | 78.7 | 83.5 | − | − | 75.0 |
| Self-Debugging Chen et al. (2024c) | gpt-3.5-turbo | 71.3 | 77.4 | − | − | − |
| | DeepSeek-Coder-6.7B-Instr | 73.2 | 77.4 | − | − | − |
| LeDex Jiang et al. (2025) | StarCoder-15B | 46.3 | 52.3 | − | − | − |
| | CodeLlama-7B | 50.0 | 55.8 | − | − | − |
| | CodeLlama-13B | 56.7 | 61.7 | − | − | − |
| CYCLE Ding et al. (2024a) | CYCLE-350M | − | 20.7 | − | − | − |
| | CYCLE-1B | − | 22.0 | − | − | − |
| | CYCLE-2.7B | − | 29.3 | − | − | − |
| | CYCLE-3B | − | 29.9 | − | − | − |
| Revisiting Self-Debugging Chen et al. (2025b) | gpt-4o | 87.8 | 92.1 | − | − | − |
| | Claude-3.5-Sonnet | 89.0 | 94.5 | − | − | − |
| | Llama-3-70B Instr. | 73.8 | 79.9 | − | − | − |
| | Qwen-2.5-Coder | 81.7 | 86.0 | − | − | − |
| SemCoder Ding et al. (2024b) | SemCoder-S-6.7B | 74.4 | 79.3 | − | − | − |
| | SemCoder-6.7B | 68.9 | 73.2 | − | − | − |

Table 11: Performance on the **HumanEval +**, **HumanEval**, **HumanEval-XL**, **HumanEval-X**, and **HumanEval-ET** benchmarks (continued). All results are reported as $pass$@1.

