# OpenReview forum: "Code Reasoning for Software Engineering Tasks: A Survey and A Call to Action"
_TMLR — Accepted by TMLR_

### Review · Reviewer_4EST · 2026-03-10

**Summary Of Contributions:**

The paper is a review of reasoning techniques for code generation and code-centric software engineering tasks. It contains a survey of the recent methodological approaches to such tasks, and performs a meta-analysis to compare their performance on several benchmarks. Beyond that, the paper also contains a comprehensive list of current gaps in the literature and calls for actions related to future directions.

**Audience:**

Yes

**Audience Explanation:**

Code generation is one of the major uses of modern LLMs. There is a huge community of both practitioners, who rely on such methods, and researchers who aim to improve these methods. I am certain both communities can benefit from such a survey, which can on one hand guide current usage of agents, and on the other hand help clarify current directions of research.

As a non-expert on the topic, I found the survey to be very interesting, and I see it as a timely addition to the literature.

**Claims And Evidence:**

Yes

**Claims Explanation:**

The paper surveys a large corpus of related of literature, which they have made available for transparency. The authors were very clear about their methodological approach to the analysis and about any potential limitations. The appendix contains a very detailed of the described results and analysis.

**Requested Changes:**

My main comment about the paper is that some parts of it, especially those describing the techniques in Section 4, can be quite dense at times. The survey introduces many different techniques and models, sometimes in only a sentence or two. This style can make it somewhat difficult to follow the main ideas. My suggestion would be to provide a somewhat broader presentation of the techniques, perhaps also with examples, possibly in an appendix. This is precisely the kind of clarification that a good survey should provide.

---

> ### Author Response · Authors · 2026-04-26
> **Author Response to Reviewer 4EST**
>
> We sincerely thank Reviewer 4EST for their valuable and encouraging assessment of our work.
>
> We appreciate the suggestion regarding the density of Section 4. We concur on the point that some techniques are introduced concisely, which can make the section challenging to follow. Including detailed examples for each method may further inflate the paper length. Due to such space constraints, these detailed examples are not included for feasibility reasons.
>
> However, to improve readability, we have included a **paper roadmap** in the appendix (pages 27-28 of the revised PDF) that outlines the structure of the paper and the key topics of each section. This will help readers navigate the content easily and locate methods of interest.
>
> We thank the reviewer again for their valuable feedback and hope this clarification addresses the concern.

---

### Review · Reviewer_81Fm · 2026-03-11

**Summary Of Contributions:**

## Summary of Contributions

This paper presents the first dedicated survey of code reasoning techniques for software engineering (SWE) tasks. The authors organize the field along two axes—techniques and tasks—and propose a taxonomy covering Chain-of-Thought (plan-based, structure-based, training-based), self-refinement (execution reflection, training with feedback, automated test generation), inference scaling (sampling, search), and SWE agents (workflow, agent optimization, training with trajectory). The survey compiles comparison tables (Tables 3–9) synthesizing reported results across common benchmarks (HumanEval, MBPP, APPS, SWE-Bench, etc.) and derives six observations about relative technique effectiveness. It also identifies eight concrete Calls to Action (CTAs) for future research. A companion GitHub repository is maintained for reproducibility.

**Key strengths:**
- Fills a genuine gap: no prior survey specifically targets inference-time code reasoning techniques for SWE tasks with a unified taxonomy.
- The taxonomy (Figure 2) and technique-approach mapping (Table 2) are well-designed and provide a useful organizing framework.
- The comparative analysis (Section 6) with Observations 1–6 is valuable, progressively building from CoT vs. self-refinement to agentic + inference scaling combinations.
- The Calls to Action (Section 7) are specific and actionable, particularly CTA-7 (error recovery benchmarks) and CTA-8 (tool-use benchmarks for agents).
- Comprehensive coverage: ~80+ papers surveyed with detailed appendix tables.

**Key weaknesses:**
- The comparative analysis relies entirely on reported numbers from different papers with different setups, limiting the reliability of cross-technique conclusions.
- The survey is heavily skewed toward code generation benchmarks; test generation, issue resolution, and code understanding receive much less analytical depth.
- Some important recent developments (e.g., Claude 3.5/4 agent capabilities, Cursor/Windsurf-style IDE agents, Devin) are not discussed despite their practical impact.

**Additional Comments:**

This is a timely and well-organized survey that fills a genuine gap in the literature. The taxonomy is the paper's strongest contribution and will be useful to the community. The main weakness is that the comparative analysis over-claims given the heterogeneous nature of the underlying results. With appropriate qualification of the claims and better scoping of the observations, this paper would make a solid contribution to TMLR.

Minor notes:
- Table 1 is a nice addition comparing survey coverage across dimensions.
- The writing is generally clear and well-structured, though some sections (e.g., 4.1) read as a catalog of papers rather than a synthesized narrative.
- Some references appear duplicated (e.g., Hendrycks et al. 2021a and 2021b appear to be the same paper with different citation entries).

**Audience:**

Yes

**Audience Explanation:**

Code reasoning for SWE tasks is a rapidly growing area at the intersection of LLM research and software engineering. The survey addresses a timely and practically important topic. Researchers working on LLM-based code generation, SWE agents, and inference-time compute would benefit from the organized taxonomy, the comprehensive reference tables, and the identification of research gaps. Practitioners building SWE tools would find the comparative tables (Tables 3–9) useful as a starting reference. The Calls to Action are also likely to inspire new research directions.

The topic is highly relevant to TMLR's audience given the journal's focus on machine learning methods and applications.

**Broader Impact Concerns:**

No significant ethical concerns. The survey itself does not introduce new methods or systems. The discussed techniques for automated code generation and SWE agents do raise general concerns about software quality assurance (automated code may contain subtle bugs or security vulnerabilities), but these are well-recognized issues in the field and not specific to this paper.

**Claims And Evidence:**

No

**Claims Explanation:**

The paper's main analytical claims (Observations 1–6) are derived from synthesizing results across different papers, each using different experimental setups, prompts, evaluation harnesses, and sometimes different benchmark versions. The authors acknowledge this limitation (Section 6, paragraph 1; Section 8), but the observations are still presented as general conclusions (e.g., "Structure-aware CoT strategies tend to outperform planning-based CoT"). Several specific concerns:

1. **Confounded comparisons.** Figures 3–7 aggregate results from papers that use different temperatures, number of samples, prompt formats, and even different splits of the same benchmark (e.g., MBPP vs. MBPP-S vs. MBPP-ET). Drawing trend conclusions across these is methodologically weak. For instance, Observation 1 compares SCoT and CGO (using gpt-3.5) against Self-Planning (using gpt-3.5-turbo and Codex), but these models differ in capability.

2. **Selective data points.** Some observations rest on a small number of comparison points. Observation 4 ("Inference scaling can also outperform self-refinement based strategies") is acknowledged to rest on only one data point. This is insufficient evidence for a generalizable claim.

3. **Missing statistical rigor.** No confidence intervals or variance analysis is provided. Many benchmarks (especially pass@1) are known to have high variance, yet results are compared at face value.

4. **Taxonomy assignment subjectivity.** Table 2 assigns each approach a "dominant" technique, but many systems are deeply hybrid. This can bias the comparative analysis—e.g., PlanSearch is categorized under Code CoT but also uses self-refinement and inference scaling, making it unclear which component drives its performance.

5. **Benchmark coverage imbalance.** The comparative discussion (Section 6) focuses almost exclusively on code generation (HumanEval, MBPP, APPS). The observations are framed generally ("on code tasks") but actually only cover one narrow task type. SWE-Bench results (Table 5) are listed but not comparatively analyzed with the same rigor.

That said, the descriptive survey content (Sections 2–5) is accurate and well-referenced, and the taxonomy itself is a valid contribution independent of the comparative claims.

**Requested Changes:**

### Critical (required for acceptance):

1. **Qualify the comparative claims.** Observations 1–6 should be explicitly framed as "trends suggested by reported results under heterogeneous conditions" rather than as general findings. Currently, phrases like "tend to outperform" imply a level of evidence that the methodology does not support. Consider adding a dedicated subsection discussing the threats to validity of the cross-paper comparison approach.

2. **Address the code-generation bias in the analysis.** Section 6 overwhelmingly focuses on code generation benchmarks. Either (a) extend the comparative analysis to cover SWE-Bench/issue resolution with the same depth (the data is already in Table 5), or (b) explicitly scope Observations 1–6 as applying to "code generation tasks" rather than "code tasks" or "SWE tasks" in general.

3. **Strengthen the evidence for weaker observations.** Observation 4 rests on a single data point comparing inference scaling and self-refinement. Either find additional evidence to support it or downgrade it to a hypothesis/open question rather than an observation.

4. **Discuss confounders more explicitly.** In each comparison figure (Figs. 3–7), note which results come from the same paper (controlled comparison) vs. different papers (uncontrolled). This would help readers assess which comparisons are more reliable.

### Suggested (would strengthen the paper):

5. **Include a discussion of commercial/industrial SWE agents.** The survey omits significant practical systems like Devin, Cursor Agent, GitHub Copilot Workspace, and similar tools that represent major real-world deployments of code reasoning. Even if proprietary, their publicly available design descriptions and benchmark results deserve mention in a comprehensive survey.

6. **Add a cost/efficiency dimension.** The current analysis focuses on accuracy (pass@k, resolved rate) but ignores computational cost. Inference scaling and agentic approaches can be orders of magnitude more expensive than single-pass CoT. A brief discussion of the accuracy-cost tradeoff would make the survey more practically useful.

7. **Expand the discussion of reasoning models.** The survey mentions DeepSeek-R1 and o1-mini in the comparison tables but does not deeply discuss how native reasoning models (with built-in chain-of-thought) change the landscape. These models blur the line between "prompting technique" and "model capability," which has implications for the taxonomy.

---

> ### Author Response · Authors · 2026-04-26
> **Author Response to Reviewer 81Fm (1/4)**
>
> We thank the Reviewer 81Fm for their very thorough and constructive comments. We address the critical points first, followed by the strengthening suggestions, and finally the specific weaknesses raised in the review.
>
> Any changes we reference below (e.g., "we have added this in Section 6") can be found in the updated PDF in red text, which signifies revisions made according to reviewer comments.
>
> > Critical 1. Qualify the comparative claims. Observations 1–6 should be explicitly framed as "trends suggested by reported results under heterogeneous conditions" rather than as general findings. Currently, phrases like "tend to outperform" imply a level of evidence that the methodology does not support. Consider adding a dedicated subsection discussing the threats to validity of the cross-paper comparison approach.
>
> We thank the reviewer for this important suggestion. We have made several changes to address this concern:
>   - **Framing of Observations.** All six observations in Section 6 have been re-worded to frame them as trends in matched reported results rather than definite claims. Each observation now begins with _"Reported results across several studies suggest that [technique] can outperform [other technique] in some code generation benchmarks,"_ replacing the prior definitive language (e.g., "outperforms," "dominates").
> - **Definitive claims removed from figure captions.** We removed all absolute comparative conclusions from the Section 6 figure captions (e.g., "Self-refinement techniques outperform code CoT based techniques"), leaving the captions to describe the experimental setup without drawing conclusions.
> - **Comparison table.** We added Table 3 in the appendix, which lists all pairwise comparisons discussed in Section 6, grouped by observation. Each row shows the two approaches, the winner, the benchmark, the model, and the margin of difference in percentage points. This makes it easy to assess both the number of matched comparisons supporting each observation and their magnitude. We note that each observation is supported by at least 8 data points across multiple base models and benchmark splits.
> - **Threats to validity subsection.** We added Section 8.1 ("Threats to Validity of Cross-Paper Comparisons") in the Limitations section, which explicitly discusses confounders such as differences in prompting strategy, inference budgets, benchmark versions (e.g., MBPP vs. MBPP-S vs. MBPP-ET), tool access, and evaluation harnesses. Our mitigation strategy is threefold: (1) we condition all comparisons on the same underlying base model, (2) we look for trends that hold consistently across multiple splits of the same benchmark and across entirely independent benchmarks, and (3) we report the margin of difference between approaches in an appendix table (Tab. 3) so readers can assess the strength of each comparison. We present suggestive claims rather than definitive conclusions and note that most papers do not report confidence intervals or variance.
> - **Framing in Introduction of Section 6.** The opening paragraph of Section 6 now explicitly states that the analysis is not exhaustive, that results are reported directly from original papers without normalization, and points readers to both the margins table and the threats-to-validity discussion.
>
> > Critical 2. Address the code-generation bias in the analysis. Section 6 overwhelmingly focuses on code generation benchmarks. Either (a) extend the comparative analysis to cover SWE-Bench/issue resolution with the same depth (the data is already in Table 5), or (b) explicitly scope Observations 1–6 as applying to "code generation tasks" rather than "code tasks" or "SWE tasks" in general.
>
> For Section 6, we found that the majority of code reasoning papers with comparable results are on code generation and APR tasks. We have added a clarification at the start of the section stating that most of its comparisons are on code generation benchmarks, with a few on issue resolution (SWE-Bench). We also explicitly adjusted the observation boxes to attribute each observation to the specific tasks it draws from.
>
> However, we believe it would not be accurate to characterize this paper as strictly a code generation paper, as many other aspects, including the taxonomy, task coverage, and call-to-action directions, cover several SWE tasks beyond code generation.

---

> > ### Author Response · Authors · 2026-04-26
> > **Author Response to Reviewer 81Fm (2/4)**
> >
> > > Critical 3. Strengthen the evidence for weaker observations. Observation 4 rests on a single data point comparing inference scaling and self-refinement. Either find additional evidence to support it or downgrade it to a hypothesis/open question rather than an observation.
> >
> > We have addressed this in two ways. First, we expanded the comparison between inference scaling and self-refinement by including additional benchmark results (MBPP, HumanEval, HumanEval+) that were already present in our tables but not discussed. The results are mixed: CodeTree outperforms Revisiting Self-Debugging on MBPP+ (80.7 vs. 76.5), MBPP (98.7 vs. 91.5), and HumanEval (94.5 vs. 92.1), but Revisiting Self-Debugging performs better on HumanEval+ (87.8 vs. 86.0). Second, we downgraded the claim: the boxed observation now states that inference scaling shows a trend of stronger performance over CoT-dominant strategies, and the text frames the inference scaling vs. self-refinement comparison as an open question, noting that the comparisons are limited to a single approach pair.
> >
> > > Critical 4. Discuss confounders more explicitly. In each comparison figure (Figs. 3–7), note which results come from the same paper (controlled comparison) vs. different papers (uncontrolled). This would help readers assess which comparisons are more reliable.
> >
> > All comparisons in Figures 3-8 are cross-paper. We have clarified this in the introduction to Section 6 and threats (8.1), which now states that each result is taken from its respective paper. We condition on the same base model and benchmark for every comparison; we never compare results across different splits of the same benchmark (e.g., a result on MBPP-S is not compared against a result on MBPP-ET, even when the base model is the same). We acknowledge limitations of uncontrolled aspects and discuss them in Section 8.1 (Threats to Validity), where we explain our mitigation strategy: looking for trends that hold across multiple benchmark splits and independent benchmarks rather than relying on any single comparison point.
> >
> >
> > > Strengthening 1. Include a discussion of commercial/industrial SWE agents. The survey omits significant practical systems like Devin, Cursor Agent, GitHub Copilot Workspace, and similar tools that represent major real-world deployments of code reasoning. Even if proprietary, their publicly available design descriptions and benchmark results deserve mention in a comprehensive survey.
> >
> > We have added a discussion in the Limitations section (Section 8) acknowledging that this survey focuses on published literature and thus excludes highly capable but closed systems such as Devin, Cursor Agent, and GitHub Copilot Workspace. Although these systems may employ code-specific reasoning, their methods are often unclear and their results difficult to verify, making them unsuitable for inclusion in a survey that aims to compare techniques on a methodological level.
> >
> > > Strengthening 2. Add a cost/efficiency dimension. The current analysis focuses on accuracy (pass@k, resolved rate) but ignores computational cost. Inference scaling and agentic approaches can be orders of magnitude more expensive than single-pass CoT. A brief discussion of the accuracy-cost tradeoff would make the survey more practically useful.
> >
> > We have added a discussion in the Limitations section (Section 8) acknowledging that cost is an important consideration, especially for test-time computation. However, most surveyed papers do not report cost-related metrics, and when they do, the reporting is inconsistent across papers, making meaningful comparison difficult.
> >
> > > Strengthening 3. Expand the discussion of reasoning models. The survey mentions DeepSeek-R1 and o1-mini in the comparison tables but does not deeply discuss how native reasoning models (with built-in chain-of-thought) change the landscape. These models blur the line between "prompting technique" and "model capability," which has implications for the taxonomy.
> >
> > We appreciate this suggestion. Though expanding the discussion to cover general reasoning models and their implications would significantly broaden the scope of the survey. This topic is already covered by other surveys (e.g., Plaat et al. 2024, Xu et al. 2025). Our focus is on code-specific reasoning techniques. Code-specific models, which have built-in reasoning are usually trained on some reasoning data, and we cover such models in the survey.

---

> > > ### Author Response · Authors · 2026-04-26
> > > **Author Response to Reviewer 81Fm (3/4)**
> > >
> > > Lastly, we would like to address some specific weaknesses raised in the review:
> > >
> > > > Weakness 1. Confounded comparisons. Figures 3–7 aggregate results from papers that use different temperatures, number of samples, prompt formats, and even different splits of the same benchmark (e.g., MBPP vs. MBPP-S vs. MBPP-ET). Drawing trend conclusions across these is methodologically weak. For instance, Observation 1 compares SCoT and CGO (using gpt-3.5) against Self-Planning (using gpt-3.5-turbo and Codex), but these models differ in capability.
> > >
> > > We appreciate the reviewer raising this concern. However, we would like to clarify that this specific example is not accurate. In Observation 1, SCoT, CGO, and Self-Planning are all compared using gpt-3.5-turbo as the base model (see Tables in the appendix). The text in the original submission shortened "gpt-3.5-turbo" to "gpt-3.5," which may have caused the confusion. We have updated the text to use the full model name "gpt-3.5-turbo" throughout to avoid ambiguity. Our comparisons are conditioned on the same base model and benchmark; we never compare results across different splits of the same benchmark (e.g., a result on MBPP-S is not compared against a result on MBPP-ET, even when the base model is the same). We have several data points across several base model-benchmark splits to support each trend, and we make this clear in the appendix table which also includes the margins.
> > >
> > >
> > > > Weakness 2. Selective data points. Some observations rest on a small number of comparison points. Observation 4 ("Inference scaling can also outperform self-refinement based strategies") is acknowledged to rest on only one data point. This is insufficient evidence for a generalizable claim.
> > >
> > > We addressed this in Critical 3.
> > >
> > > > Weakness 3. Missing statistical rigor. No confidence intervals or variance analysis is provided. Many benchmarks (especially pass@1) are known to have high variance, yet results are compared at face value.
> > >
> > > We address this in Section 8.1, where we specify we cannot add confidence intervals or variance analysis as these are cross-paper comparisons and the original papers do not consistently report them. To mitigate this, we report all supporting data points and margins in the appendix table, allowing readers to judge whether differences are meaningful or marginal, and look for trends that hold across multiple data points rather than relying on any single comparison point. We note that each observation is supported by at least 8 data points across multiple base models and benchmark splits, as detailed in the appendix margins table.
> > >
> > >
> > > > Weakness 4. Taxonomy assignment subjectivity. Table 2 assigns each approach a "dominant" technique, but many systems are deeply hybrid. This can bias the comparative analysis—e.g., PlanSearch is categorized under Code CoT but also uses self-refinement and inference scaling, making it unclear which component drives its performance.
> > >
> > > We appreciate this concern. We have added clarification in both the introduction to Section 4 and in the Limitations section (Section 8). Specifically, we describe the decision process used to assign dominant techniques: each author independently reviewed their assigned papers and determined the dominant technique by asking two guiding questions: (1) What is the core method, novelty, or contribution of the paper? (2) Which technique, if excluded, would cause the paper to fall outside the scope of this survey? This grounds the labeling in the paper's primary contribution rather than any incidental technique it happens to use.
> > >
> > > For example, MSCoT extends SCoT to multiple languages and employs both structured CoT and training with CoT. However, its primary contribution is generating CoT data for training, not the structural decomposition itself, so its dominant technique is assigned as "Training with CoT."
> > > Similarly, SemCoder leverages generated tests and self-refinement, but its core method is training a model to reason about program semantics, placing it under training-based CoT rather than self-refinement.
> > >
> > > Table 2 also lists secondary techniques for each approach, so hybrid methods are represented beyond their dominant label.

---

> ### Author Response · Authors · 2026-04-26
> **Author Response to Reviewer 81Fm (4/4)**
>
> > Weakness 5. Benchmark coverage imbalance. The comparative discussion (Section 6) focuses almost exclusively on code generation (HumanEval, MBPP, APPS). The observations are framed generally ("on code tasks") but actually only cover one narrow task type. SWE-Bench results (Table 5) are listed but not comparatively analyzed with the same rigor.
>
> We address this in Critical 2.
>
> We thank the reviewer again for their valuable feedback, which has meaningfully improved the paper. We hope we have satisfactorily addressed the concerns raised.

---

### Review · Reviewer_trRS · 2026-04-12

**Summary Of Contributions:**

This paper surveys reasoning methods for code and software engineering tasks, with emphasis on test-time/inference-time methods. It proposes a taxonomy spanning code CoT, self-refinement, inference scaling, and SWE agents; catalogs tasks and benchmarks across code generation, test generation, issue resolution, and reasoning/understanding; and adds a cross-paper comparison section plus a set of “call to action” future directions. The paper also positions itself as the first dedicated survey of code reasoning for SWE tasks.

Key strengths:

* The topic is timely and important. The intersection of reasoning, code LLMs, and agentic SWE is moving quickly, and a focused synthesis is valuable.
* The taxonomy is useful and easy to understand at a high level, especially the distinction between prompting-style reasoning, execution-feedback methods, search/inference scaling, and agentic scaffolding.
* The benchmark catalog and discussion of under-explored evaluation settings are helpful, and the future directions are generally thoughtful and relevant.

Key weaknesses:

* The strongest “Observation 1–6” claims in Section 6 are only partially supported, because the evidence is synthesized across heterogeneous papers without normalizing prompts, compute budgets, tool access, or evaluation harnesses. The manuscript acknowledges this limitation, but the resulting conclusions are still stated too strongly.
* The taxonomy and comparative analysis rely on a “dominant technique” assignment even though many methods are hybrids and the paper itself notes that this categorization is subjective. That makes causal attribution difficult.
* There appear to be figure/metric inconsistencies that undermine confidence in the comparative section. Most notably, Figure 4’s CodeContests panel seems to use numbers marked as pass@5 in Table 4 while the figure axis says pass@1, and Figure 7 uses a pass@1 axis while one panel is SWE-Bench Lite, whose table reports resolved rate.

**Additional Comments:**

I found the paper readable, timely, and potentially quite useful. The taxonomy and benchmark coverage are the main strengths. The biggest thing keeping this from being a strong survey paper, in my view, is that the comparative section currently claims more than the underlying cross-paper evidence can really support. If the authors tighten Section 6, fix the figure/metric inconsistencies, and make the survey methodology more reproducible, I would be substantially more enthusiastic.

**Audience:**

Yes

**Audience Explanation:**

This is a timely topic at the intersection of LLM reasoning, code generation, agentic systems, evaluation, and software engineering. Researchers entering this area would likely benefit from the taxonomy, the benchmark map, the summary of hybrid systems, and the discussion of open problems. Even readers who do not work directly on SWE agents may find the comparison between structure-aware prompting, execution feedback, and search-based inference scaling useful.

More broadly, TMLR explicitly includes surveys that draw new connections, highlight trends, and suggest new problems. This paper is clearly trying to do exactly that, and I think at least a meaningful slice of TMLR’s audience would want such a reference point.

**Broader Impact Concerns:**

I do not see a major ethics blocker here, since this is a survey rather than a release of a new system. That said, I would welcome a brief broader-impact discussion if one is not already present. The paper explicitly argues that more SWE tasks are likely to be automated, and it also mentions security as a desirable evaluation dimension in future work. That creates at least mild dual-use and deployment concerns: misuse for malicious code generation or vulnerability exploitation, over-trust in autonomous code repair, and labor/displacement effects for software developers. A short paragraph acknowledging these issues and emphasizing responsible benchmarking/deployment would be sufficient in my view.

**Claims And Evidence:**

No

**Claims Explanation:**

**Partially.**

The descriptive and organizational claims are mostly supported. The paper does assemble a broad recent literature, situates itself relative to adjacent surveys, defines a clear taxonomy, and is transparent that its comparative section is based on reported numbers rather than a controlled re-evaluation. Those parts are useful and, in my view, largely convincing.

Where I become less convinced is in the stronger comparative claims. Section 6 explicitly says the comparison is cross-paper, restricted to overlapping model/benchmark intersections, and not normalized for prompts, compute budgets, tool access, or evaluation details. Yet the paper then states fairly strong observations such as “X outperforms Y” and attributes the gains to categories like inference scaling or agentic scaffolding. Because many of the methods compared are hybrids, and because compute/tool budgets are central confounders in this literature, these observations read more strongly than the underlying evidence warrants. I would be more comfortable if the claims were framed as *associations seen in matched reported results* rather than as general superiority statements.

I also think there are concrete presentation issues that need correction. Figure 4 is labeled with a pass@1 axis, but the CodeContests numbers it appears to plot correspond to Table 4 entries that are flagged as pass@5 for CodeChain and AlphaCodium. Figure 7 likewise uses a pass@1 y-axis while including SWE-Bench Lite, whose table uses resolved rate. These are not merely cosmetic issues: they directly affect how readers interpret the comparative evidence.

So my bottom line is: the survey component is strong enough to be useful, but the evidence behind the paper’s headline comparative observations is only moderately convincing in its current form.

**Requested Changes:**

1. **[Critical] Rework the tone and methodology of Section 6.**
   The current comparison section should either become more systematic or become more cautious. At minimum, each observation should be clearly framed as a trend in matched reported results rather than a broad superiority claim. Ideally, the authors would report how many matched comparisons support each observation, and make compute/tool-budget caveats more prominent near the figures themselves, not only in the limitations section.

2. **[Critical] Fix the metric inconsistencies in the figures.**
   Figure 4 appears inconsistent with Table 4 for CodeContests: the figure uses a pass@1 axis, while the relevant Table 4 entries are marked pass@5. Figure 7 uses a pass@1 axis while including SWE-Bench Lite, which is reported in resolved rate. These should be corrected before publication. Either separate metrics by panel or use task-specific axis labels/captions.

3. **[Critical] Strengthen the survey methodology section.**
   Section 2 is currently too light for a survey paper. Please add exact search dates, concrete query strings, inclusion/exclusion criteria, approximate numbers of papers screened and retained, and the rule for how the living GitHub list relates to the static snapshot in the paper. A short PRISMA-style flow figure or appendix table would help substantially.

4. **[Critical] Clarify the scope boundary.**
   The paper says purely training-time techniques are out of scope, but the taxonomy includes training with CoT, training with feedback, and training with trajectory. This is not necessarily wrong, but the operational inclusion rule needs to be made explicit. Otherwise the paper’s scope reads as broader than advertised.

5. **[Critical] Clarify how “dominant technique” is assigned, or revise the taxonomy presentation.**
   Since many methods are hybrids and the paper itself acknowledges that dominant-technique assignment is subjective, please either provide explicit decision rules for this assignment or consider a representation where agentic scaffolding is orthogonal to the underlying reasoning method. This would also make Section 6’s attributions more credible.

6. **[Critical] Narrow or better substantiate the novelty claim.**
   The claim that this is the “first” dedicated survey in this space may be correct, but it currently rests on a somewhat subjective coverage matrix. Please either soften that claim or make the criteria for Table 1 more explicit and reproducible, especially relative to overlapping recent surveys cited in the manuscript.

7. **[Strengthening] Better separate evidence-backed conclusions from speculative hypotheses in Section 7.**
   The CTA section is interesting, but some of its theoretical explanations feel more conjectural than directly established by the literature synthesis. I would recommend labeling those parts more explicitly as hypotheses or interpretations.

8. **[Strengthening] Either broaden the comparative analysis beyond code generation or narrow the rhetoric.**
   Most of the concrete comparisons are still about code generation, with relatively limited direct evidence for broader SWE tasks. I would either add more systematic comparisons for issue resolution/test generation, or more clearly state that the empirical comparison findings in Section 6 mainly concern code generation.

9. **[Strengthening] Distinguish peer-reviewed papers from preprints.**
   Since many cited works are recent arXiv papers, a column or visual marker indicating publication status would improve interpretability for readers.

10. **[Strengthening] Fix copyediting and section-reference issues.**
    I noticed at least two likely reference errors: “Self-refinement (Sec. 5.4)” appears to mean Sec. 4.2, and “Appendix 2” appears to mean Section 2. There are also some notation/readability issues (e.g., the q/✓/¥ symbols and a few capitalization/grammar problems) that would benefit from a final cleanup.

---

> ### Author Response · Authors · 2026-04-26
> **Author Response to Reviewer trRS (1/2)**
>
> We thank Reviewer trRS for their very thorough and constructive comments. We address the critical points first, followed by the strengthening points.
>
> Any changes we reference below (e.g., "we have added this in Section 6") can be found in the updated PDF in **red** text, which signifies revisions made according to reviewer comments.
>
> > 1. [Critical] Rework the tone and methodology of Section 6.
>
> We thank the reviewer for this important suggestion. We have made several changes to address this concern:
>   - **Framing of Observations.** All six observations in Section 6 have been re-worded to frame them as trends in matched reported results rather than definite claims. Each observation now begins with _"Reported results across several studies suggest that [technique] can outperform [other technique] in some code generation benchmarks,"_ replacing the prior definitive language (e.g., "outperforms," "dominates").
> - **Definitive claims removed from figure captions.** We removed all absolute comparative conclusions from the Section 6 figure captions (e.g., "Self-refinement techniques outperform code CoT based techniques"), leaving the captions to describe the experimental setup without drawing conclusions.
> - **Comparison table.** We added Table 3 in the appendix, which lists all pairwise comparisons discussed in Section 6, grouped by observation. Each row shows the two approaches, the winner, the benchmark, the model, and the margin of difference in percentage points. This makes it easy to assess both the number of matched comparisons supporting each observation and their magnitude. We note that each observation is supported by at least 8 data points across multiple base models and benchmark splits
> - **Threats to validity subsection.** We added Section 8.1 ("Threats to Validity of Cross-Paper Comparisons") in the Limitations section, which explicitly discusses confounders such as differences in prompting strategy, inference budgets, benchmark versions (e.g., MBPP vs. MBPP-S vs. MBPP-ET), tool access, and evaluation harnesses. Our mitigation strategy is threefold: (1) we condition all comparisons on the same underlying base model, (2) we look for trends that hold consistently across multiple splits of the same benchmark and across entirely independent benchmarks, and (3) we report the margin of difference between approaches in an appendix table (Tab. 3) so readers can assess the strength of each comparison. We present suggestive claims rather than definitive conclusions and note that most papers do not report confidence intervals or variance. The Comparison and Discussion section (Sec. 6) now explicitly points readers to this threats to validity discussion.
> - **Framing in Introduction of Section 6.** The opening paragraph of Section 6 now explicitly states that the analysis is not exhaustive, that results are reported directly from original papers without normalization, and points readers to both the margins table and the threats-to-validity discussion.
>
> > 2. [Critical] Fix the metric inconsistencies in the figures
>
>  We have addressed this as follows:
>
> - Figure 4 (CodeContests): The CodeContests subplot now has its own y-axis label "Pass@5 (%)" to reflect that the plotted results (CodeChain and AlphaCodium) are reported as pass@5 in Table 6 of the appendix. The caption now also notes that CodeContests results are reported as pass@5.
>
> - Figure 8 (SWE-Bench Lite): The SWE-Bench Lite subplot now uses a separate y-axis label "Resolved (%)" to distinguish it from the pass@1 metric used by the code generation subplots. The caption explicitly notes which benchmarks use pass@1 and which use resolved rate.
>
> > 3. [Critical] Strengthen the survey methodology section.
>
> We have included a PRISMA-style flow diagram (Figure 2) and added more details about the search process, including search terms, inclusion criteria, and screening steps in Section 2. Also in Section 2, we clarify that new papers included will be categorized according to the Code Reasoning Taxonomy.
>
> > 4. [Critical] Clarify the scope boundary.
>
> Training is included when it is specifically designed to teach or improve an inference-time reasoning method covered by our taxonomy, such as generating CoT traces, self-refining with execution feedback, or replicating agent reasoning workflows. We did not find any pre-training papers which met this criterion. Any other training aspect of these papers is a secondary side-effect of the paper's core contributions and is not the focus of our survey. This clarification has been added in Section 2 (Survey Methodology).

---

> > ### Author Response · Authors · 2026-04-26
> > **Author Response to Reviewer trRS (2/2)**
> >
> > > 5. [Critical] Clarify how “dominant technique” is assigned, or revise the taxonomy presentation
> >
> > We appreciate this concern. We have added clarification in both the introduction to Section 4 and in the Limitations section (Section 8). Specifically, we describe the decision process used to assign dominant techniques: each author independently reviewed their assigned papers and determined the dominant technique by asking two guiding questions: (1) What is the core method, novelty, or contribution of the paper? (2) Which technique, if excluded, would cause the paper to fall outside the scope of this survey? This grounds the labeling in the paper's primary contribution rather than any incidental technique it happens to use.
> >
> > For example, MSCoT extends SCoT to multiple languages and employs both structured CoT and training with CoT. However, its primary contribution is generating CoT data for training, not the structural decomposition itself, so its dominant technique is assigned as "Training with CoT." Similarly, SemCoder leverages generated tests and self-refinement, but its core method is training a model to reason about program semantics, placing it under training-based CoT rather than self-refinement.
> >
> > Table 2 also lists secondary techniques for each approach, so hybrid methods are represented beyond their dominant label.
> >
> > > 6. [Critical] Narrow or better substantiate the novelty claim.
> >
> > We have added a new appendix table (Table 4) that provides explicit justifications for every coverage scenario in Table 1. For each survey and each dimension (Reasoning, SWE Tasks, Agents, Taxonomy, Benchmarks), the table explains why it received its rating (not covered, partial, or comprehensive). For example, Qiao et al. (2022) receives comprehensive coverage for Reasoning because it extensively studies CoT and prompting approaches, but receives "not covered" for all other dimensions because its taxonomies and benchmarks target general reasoning (arithmetic, commonsense) rather than code. Similarly, Sun et al. (2024) receives the comprehensive ratings across Reasoning and SWE Tasks but only partial for Benchmarks, as it does not offer the depth of a dedicated benchmark survey. This table allows readers to independently assess and understand the rationale behind each categorization in Table 1 and understand the specific basis for our novelty positioning.
> >
> > > 7. [Strengthening] Better separate evidence-backed conclusions from speculative hypotheses in Section 7.
> >
> > We have addressed this by identifying the two CTAs (CTA-1 and CTA-4) that stem from our own interpretive reasoning rather than established findings; we mark them with a red star symbol (⋆). The associated footnote clarifies: _"This CTA stems from hypotheses we formulate based on general trends observed in our analysis, and should be treated as interpretive hypotheses rather than definitive conclusions."_ The remaining CTAs (2, 3, 5, 6, 7, 8) are straightforward evidence-backed research gaps identified directly from the literature and are left unmarked.
> >
> > > 8. [Strengthening] Either broaden the comparative analysis beyond code generation or narrow the rhetoric.
> >
> > For Section 6, we found that the majority of code reasoning papers with comparable results are on code generation and APR tasks. We have added a clarification at the start of the section stating that most of its comparisons are on code generation benchmarks, with a few on issue resolution (SWE-Bench). We also explicitly adjusted the observation boxes to attribute each observation to the specific tasks it draws from.
> >
> > However, we believe it would not be accurate to characterize this paper as strictly a code generation paper, as many other aspects, including the taxonomy, task coverage, and call-to-action directions, cover several SWE tasks beyond code generation.
> >
> > > 9. [Strengthening] Distinguish peer-reviewed papers from preprints.
> >
> > We have addressed this by distinguishing pre-prints from publications with a gamma (γ) symbol in the taxonomy figure (Figure 3). The symbol is also defined in the figure caption.
> >
> > > 10. [Strengthening] Fix copyediting and section-reference issues.
> >
> > We have fixed the two reference errors and have tried our best to address all capitalization and grammar issues. Regarding the symbol rendering issues (q/✓/¥), we were unable to locate the .tex file and in the compiled PDF with the search functionality. It is possible that this is a rendering issue specific to the reviewer’s PDF viewer, particularly in how checkmarks are displayed in the hybrid table. Please let us know if you still see any issues.
> >
> > We thank the reviewer again for their valuable feedback, which has meaningfully improved the paper. We hope we have satisfactorily addressed the concerns raised.

---

> ### Comment · Reviewer_trRS · 2026-05-13
>
> Thank you for the detailed rebuttal and revision. I appreciate the substantial effort you put into addressing my concerns, and I believe the manuscript has improved meaningfully.
>
> In particular, I appreciate the more careful framing of Section 6, the addition of the threats-to-validity discussion and comparison table, the corrections to the figure/metric labeling, the improved survey methodology section, and the clearer discussion of dominant versus hybrid techniques. These changes address the main reasons for my earlier hesitation.
>
> I still have a few minor residual comments, but they are no longer acceptance-critical. First, some surrounding prose in Sections 6–7 remains slightly stronger than the more cautious framing used in the observation boxes, so I would recommend one final editorial pass to align the wording throughout. Second, the methodology would be even stronger if the final version included exact search dates and, if feasible, the exact query strings. Third, I noticed a remaining category-label inconsistency around PlanSearch between Table 2 and Table 3 that should be cleaned up.
>
> Overall, I am satisfied that the revision has addressed my main concerns.

---

> > ### Author Response · Authors · 2026-05-13
> >
> > Thank you for the thoughtful response and for recognizing the changes in the revision. We appreciate your assessment that the main concerns have been addressed. We will make a final editorial pass to align the prose in Sections 6 and 7 with the more cautious framing, update with more precise search dates and query strings where feasible, and correct the remaining PlanSearch category-label inconsistency in the next revision.

---

### Decision · Action_Editor_t76H · 2026-05-13

**Recommendation:** Accept with minor revision

**Additional Comments:**

I recommend acceptance with minor revision. The submission has improved substantially after revision, and the main acceptance-critical concerns have been addressed. The authors have softened the comparative claims, improved the methodology description, added a threats-to-validity discussion, corrected metric/figure inconsistencies, clarified the scope boundary, and better explained the dominant-technique assignment.

The remaining changes are minor and should be addressed in the final version: (1) perform one final editorial pass to ensure that the prose in Sections 6-7 consistently reflects the cautious framing used in the observation boxes; (2) include exact search dates and, if feasible, exact query strings for the survey methodology; and (3) clean up the remaining category-label inconsistency around PlanSearch noted by one reviewer.

**Audience:**

Yes

**Audience Explanation:**

Yes. The paper addresses a timely topic at the intersection of code reasoning, large language models, software engineering agents, inference-time computation, and evaluation. The taxonomy, benchmark organization, comparative synthesis, and discussion of open challenges are likely to be useful to researchers and practitioners working on LLM-based code generation, software engineering automation, agentic systems, and test-time reasoning. All reviewers agree that the paper would be of interest to at least some part of the TMLR audience.

**Claims And Evidence:**

Yes

**Claims Explanation:**

Yes. The revised manuscript has addressed the main concerns raised by the reviewers. In particular, the authors substantially softened the comparative claims in Section 6, clarified that the cross-paper comparisons should be interpreted as trends under heterogeneous conditions rather than controlled causal conclusions, added a dedicated threats-to-validity discussion, improved the survey methodology, and corrected the key figure/metric inconsistencies. The revised version also better clarifies the scope of the survey and the assignment of dominant versus secondary techniques.

All three reviewers now answer Yes to the claims-and-evidence question. The remaining concerns are minor and mostly editorial, such as further aligning the wording in Sections 6-7 with the more cautious framing, adding exact search dates/query strings if feasible, and cleaning up small taxonomy-label inconsistencies. These issues do not undermine the overall soundness of the submission.

---

> ### Author Response · Authors · 2026-06-18
> **Camera-ready revision update**
>
> Dear Action Editor and Reviewers,
>
> We have uploaded the camera-ready revision addressing the minor-revision requests.
>
> In particular, we made a final editorial pass over Sections 6–7 to align the prose with the cautious framing used in the observation boxes, including removing remaining broad generalizations and clarifying that the comparative claims refer to the benchmarks studied in the survey. We also added more precise search dates to the survey methodology, clarified that the initial search was conducted from February to May 2025 with a further update in December 2025, and cleaned up the remaining PlanSearch category-label inconsistency in Table 3 with an added footnote clarification.
>
> We also corrected minor typos, internal section references, and figure-caption formatting issues.
>
> Thank you again for the constructive feedback throughout the review process.